# Stabilized real-time Brillouin microscopy reveals fractal organization of protein condensates in living cells

Claudia Testi [1] ✉, Emanuele Pontecorvo[1,2], Chiara Bartoli[1], Chiara Marzaro[1], Fabrizio Gala [1,2], Li Zhang[1], Giulia Zanini [1,2], Noemi D'Abbondanza[1], Maria Giovanna Garone [3], Valeria de Turris [1], Andrea Giuliani [4], Gaia Di Timoteo [4], Irene Bozzoni [1,4], Alessandro Rosa [1,5] & Giancarlo Ruocco [1,6]

Mechanical alterations of protein condensates are increasingly recognized in the etiology of several neurodegenerative diseases, yet their characterization remains technically challenging. Although Brillouin microscopy could offer a promising solution, its use is hindered by instrumental instabilities demanding frequent adjustments and manual calibrations with reference materials. Here, we present an enhanced Brillouin Microscope that incorporates an electro-optic modulator, serving simultaneously as frequency reference, spectro-meter calibrator, and temporal stabilizer. This integration enables robust, real-time spectral stability over multiple days in a fully automated workflow. Using this system, we quantify Brillouin shifts of several protein condensates in living cells and validate our findings with FRAP. The correlation between techniques reveals a fractal internal architecture of the condensates, providing important insights into their physical nature while probing the mechanical behavior of entire compartments containing multiple protein species. Our method offers a unique framework for distinguishing physiological from pathological condensates, paving the way for long-term, user-independent, high-precision mechanical measurements in living cells.

Mechanotransduction is the process by which cells compartments sense and respond to external mechanical stimuli, converting physical forces into biochemical signals through complex cellular pathways[1]. Changes in the mechanical properties of cells and tissues are increasingly recognized as hallmarks of various pathological conditions: in this context, investigating mechanical alterations of biomolecular condensates, such as stress granules (SGs), may offer key insights into the mechanisms underlying several neurodegenerative diseases[2]. SGs are small (~1-3 μm diameter) membrane-less cytoplasmic compartments that are physiologically formed by cells in response to stressors such as oxygen reactive species, heat shock or toxin exposure[3,4]. They consist of a dynamic network of protein–protein, protein–RNA, and RNA–RNA interactions[4]. These condensates, formed via liquid-liquid phase separation (LLPS[5–9]), assemble dynamically during stress and spontaneously dissolve upon stress resolution[10–12]. SGs function as hubs for biochemical reactions and sequestration of specific molecular components. Fine-tuning of SGs mechanical properties is critical for these roles[2,4,9,13,14]:

[1]Center for Life Nano- and Neuro-Science, Istituto Italiano di Tecnologia, Rome, Italy. [2]CrestOptics S.p.A, Rome, Italy. [3]Stem Cell Medicine Department, Murdoch Children's Research Institute, Parkville, VIC, Australia. [4]Department of Biology and Biotechnology Charles Darwin, Sapienza University of Rome, Rome, Italy. [5]Department of Molecular Medicine, Sapienza University of Rome, Rome, Italy. [6]Dipartimento Di Fisica, Sapienza University of Rome, Rome, Italy. ✉e-mail: claudia.testi@iit.it

dysregulation of phase transitions, due to mutations or chronic stress, triggers SGs transition in cells from a liquid-like to a solid/gel-like state through a liquid-to-solid phase transition (LSPT[2,4,10,15]), more specifically described as a gelation process[13]. According to many recent studies[2,4,16], this pathological transition would result in the insoluble cytoplasmic aggregates characteristic of post-mortem tissues from patients suffering from neurodegenerative diseases such as frontotemporal dementia (FTD) and amyotrophic lateral sclerosis (ALS). The mechanical characterization of SGs is thus crucial, as their liquid-like state is reversible and functional, while their gel-like form is associated with toxicity and aggregation[2,4].

Despite their biological significance, however, the biophysical mechanisms underlying the LLPS-to-LSPT shift remain poorly understood, largely due to limitations of current techniques in measuring mechanical properties in living cells with sufficient spatial and temporal resolution[2,5,15,17]. Indeed, SGs small size and mobility require a spatiotemporal resolution similar to confocal microscopy, which many techniques lack; additionally, conventional biomechanics methods rely on physical contact with the sample, rendering them incompatible with in vivo and 3D biological environments[5,17–20]. Fluorescence Recovery After Photobleaching (FRAP) remains one of the few standard methods widely used *in cellulo* to infer mechanical properties from the measurement of SG components mobility, but it requires a fluorescent tag and it can't generate spatially resolved maps inside the specimens[13,18,21]. As a result, in mechanobiology research there is a growing demand for non-invasive and label-free techniques capable of probing SGs mechanical properties with the adequate resolution in living cells.

Brillouin Microscopy could be used for such a purpose. This is a novel optical technique that enables the measurement of a material's mechanical properties at the sub-micron scale in a non-invasive, label-free and non-contact way; it also integrates easily with confocal fluorescence microscopy[19,20,22–24]. This imaging technique provides a complete characterization of the viscoelastic properties of a sample through the analysis of its point-by-point Brillouin scattering spectrum. This spectrum (Fig. 1A, left panel) consists of a main central elastic peak (Rayleigh) at the incident light frequency, and two symmetric side peaks (Brillouin peaks), which arise from the inelastic scattering of light at lower (Stokes) and higher (anti-Stokes) frequencies upon interaction with the sample's thermally activated sound waves. The Brillouin peaks are characterized by the shift $\nu_B$ and a full width at half maximum (FWHM) $\Gamma_B$. From this spectrum it is possible to determine the complex longitudinal modulus M of the sample, which is composed of a real part (M'), related to the elastic response, and an imaginary part (M''), related to the dissipative response of the material[20]. Modern Brillouin Microscopes (BMs) used in biomechanical imaging require interferometric spectrometers which are typically based on VIPA (Virtually Imaged Phase Array) etalons[20] that spatially separates the stronger Rayleigh signal from the much weaker Brillouin: a Brillouin triplet acquired with a VIPA is thus repeated through different dispersion orders, separated by its Free Spectral Range (FSR), resulting in a spectrum as in Fig. 1A, right panel.

The application of Brillouin microscopy in the life sciences is rapidly expanding, enabling the assessment of biomechanical properties from subcellular compartments[20,24–28] to entire tissues[29–33] under both physiological and pathological conditions, underscoring its diagnostic potential[34–36]. However, a key limitation hindering the broader use of this technique in biological contexts is its susceptibility to temporal instabilities. Brillouin datasets are often affected by spectral drifts over time, primarily attributed to unavoidable ambient temperature fluctuations, which compromise the frequency stability of the laser source or the spectrometer[20,23]. As shown in Supplementary Fig. 1A, such drifts were observed in our standard BM equipped with a free-running 532 nm laser during time-lapse measurements on a water sample. These instabilities persisted even when using a frequency-locked laser (Supplementary Fig. 1B), indicating that the source of the drift is not limited to the laser itself, but also includes thermal expansion of the interferometer cavity or other optomechanical components of the spectrometer[37]. Consequently, both Brillouin frequency shift and linewidth of our water datasets exhibited substantial temporal variations, with $\nu_B$ fluctuating by approximately 1%.

Another major limitation of current state-of-the-art BMs lies in the calibration of the spectrometer from pixel to frequency units. The currently accepted protocol, as reported in the literature[20,23], involves utilizing the known Brillouin frequency shifts of standard reference materials -such as water and methanol- to fit the frequency dispersion curve of the VIPA etalon (an example is provided in Supplementary Fig. 1C). However, this method is highly sensitive to factors that are independent of the spectrometer's alignment. Specifically, the Brillouin shift is known to vary with external parameters such as temperature[38] and the choice of microscope objective used[39], as illustrated in Supplementary Fig. 2.

These issues can substantially impact the performance of BMs in biological applications, raising concerns regarding the stability and reproducibility of Brillouin measurements. This is particularly critical, given that the variation in the Brillouin frequency shift across different cellular regions typically falls within a narrow range. For this reason, state-of-the-art BMs designed for biological investigations are expected to achieve spectral precision of ≤10 MHz, corresponding to a stability on $\nu_B$ of ~0.1%[20,23]. However, maintaining such a high level of stability is technically demanding in the presence of drifts: for instance, as shown in the water dataset presented in Supplementary Figs. 1A and B, spectral shifts exceeded the required precision of $\nu_B$ by nearly an order of magnitude.

Such temporal instabilities compromise the quality and reliability of Brillouin maps of cells, where acquisitions usually require several minutes. For instance, Fig. 1B presents fluorescence, brightfield, and Brillouin data of a SK-N-BE cell overexpressing TAR DNA-binding protein (TDP-43) localized in cytoplasmic condensates. Brillouin imaging was affected by pronounced spectral drifts, which either distorted the spectra (orange trace, right panel) or caused Rayleigh line spillover (yellow trace), leading to signal saturation and rendering the dataset unusable. The current workaround to this problem, as commonly reported in the literature[20,23], involves frequent (approximately every 10–30 min) manual realignments of the spectrometer and repeated calibrations over time, as illustrated in the lower schematic of Fig. 1B. This procedure ensures that the Brillouin spectrum remains centered and that the calibration remains valid during time, but it is labor-intensive, requires continuous user intervention, and limits the applicability of Brillouin microscopy only to static, short-term experiments. As a result, it poses a significant barrier to automated, long-term studies such as time-lapse imaging. Moreover, even with frequent calibrations, another recurring issue is the day-to-day variability in Brillouin shift measurements, as shown in Supplementary Fig. 1D: here, the apparent differences observed between Brillouin maps are not attributable to genuine sample changes but rather to instrument-induced artifacts, undermining the reliability of long-term measurements.

A recent consensus paper on the use of Brillouin microscopy for biological materials[20] emphasized the urgent need for robust strategies to ensure that Brillouin data acquired from biological samples are comparable across experimental conditions and laboratories. However, when water and methanol are used as calibration standards, no universally accepted absolute reference values are available for this critical step; furthermore, no current strategies effectively compensate for thermal drifts. These limitations significantly reduce the accuracy and repeatability of Brillouin measurements. Consequently, the application of Brillouin microscopy to research areas such as the characterization of biomolecular condensates has been largely

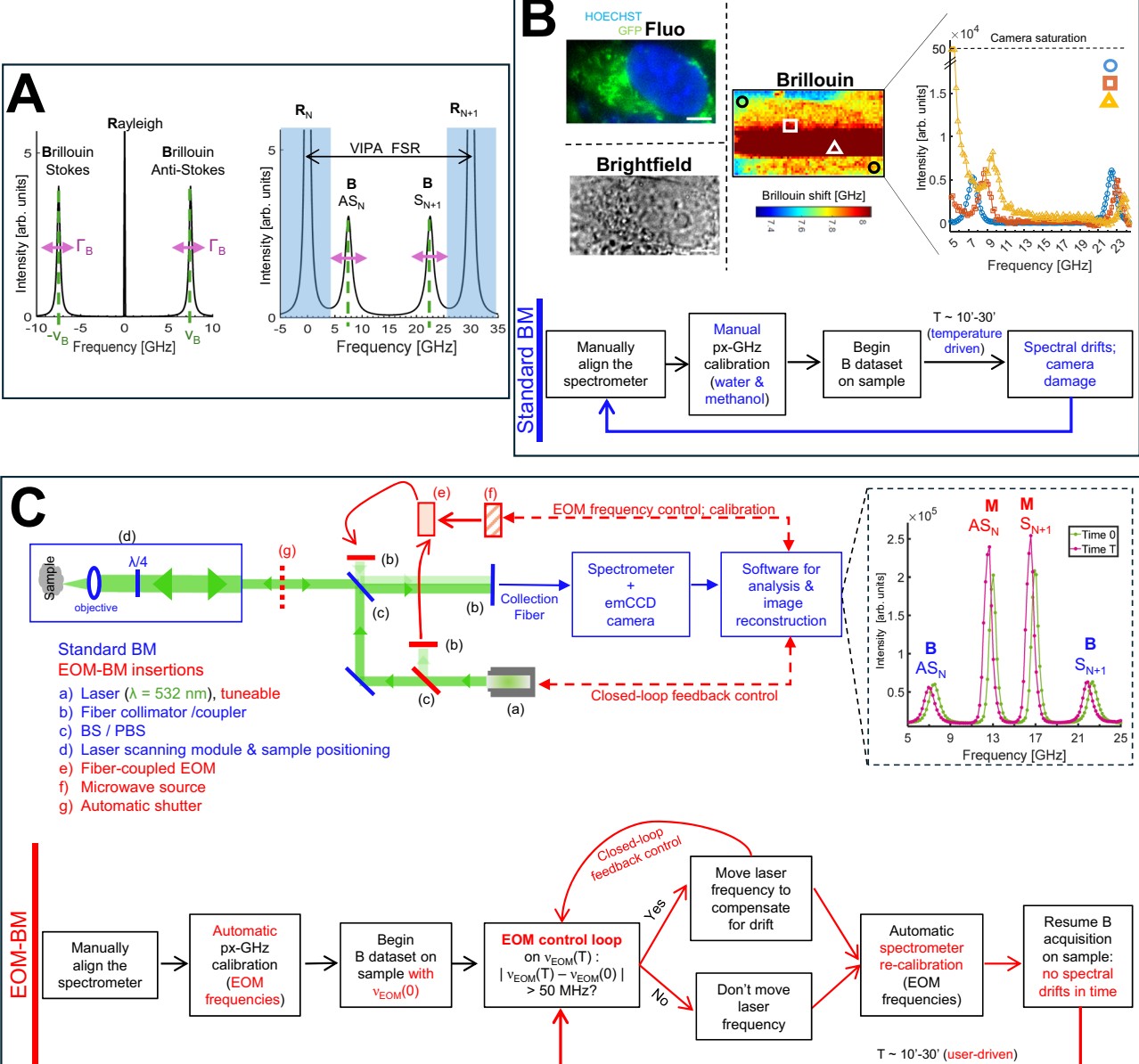

**Fig. 1 | Implementation of an Electro-Optic Modulator (EOM) source in an existing standard Brillouin Microscope (BM). A** Left panel: Brillouin spectrum of a material, characterized by a frequency shift ($\nu_B$, green dotted lines) and linewidth ($\Gamma_B$, violet arrows). Right panel: Brillouin spectrum detected through a VIPA. $AS_N$: Anti-Stokes Brillouin peak, dispersion order N; $S_{N+1}$: Stokes Brillouin peak, dispersion order N + 1; FSR: Free Spectral Range. In a standard double-VIPA BM, Rayleigh signals are blocked (shaded blue rectangles) and only the Brillouin peaks are acquired. **B** Upper panel: SK-N-BE cell imaged with fluorescence (HOECHST-stained nucleus in blue, GFP tagged TDP-43 protein in green), brightfield and Brillouin microscopy. Brillouin spectra (right) show the presence of drifts that irremediably alter the data (red stripes in the image, i.e., yellow and orange Brillouin spectra on the right) and cause Rayleigh spillover, saturating camera signal. Scale bar = 5 μm. These experiments were repeated several times with consistent results. Lower panel: data acquisition workflow in a standard BM, based on frequent and manual realignments of the spectrometer[20,23] every T ~ 10−30 min. **C** Upper panel: standard

BM (blue) equipped with our EOM-BM insertions (red). Inputs to the EOM (element e) are the laser signal (a) and a microwave source (f); its output is an optical signal shifted in frequency and injected into the collection fiber together with sample Brillouin signal (from d). The setup is governed by an additional custom-developed software that governs EOM frequency, performs automatic pixel-GHz calibrations and executes the closed-loop feedback control. Right dotted insert: water spectra obtained with our EOM-BM. (B): Brillouin signals from the sample; (M): EOM Anti-Stokes and Stokes signals, at $\nu_{EOM}$ and FSR-$\nu_{EOM}$ respectively (here, $\nu_{EOM} = 13$ GHz). EOM peaks act as a reference, replacing the Rayleigh signals and allowing to quantify temporal drifts (green and pink spectra, acquired at T = 5' difference). Lower panel: data acquisition workflow of our EOM-BM setup: the system is calibrated with known frequencies given by the EOM; the EOM control loop automatically compensates for thermal drifts every T ~ 10−30 min, replacing manual adjustments. This ensured that Brillouin data remained stable for days in a user-independent workflow. Source data are provided as Source Data file.

restricted to in-vitro systems[40] or fixed samples[41,42], and even in these contexts, discrepancies in results across different laboratories have raised important concerns regarding the reliability of Brillouin microscopy for such sensitive biological investigations.

 In this work, we addressed the challenges of instrumental instability and sample-dependent calibration by integrating an Electro-

Optic Modulator (EOM)[43] into a state-of-the-art Brillouin microscope. The EOM simultaneously functioned as an internal frequency reference within each measurement and as a calibration standard for the spectrometer. Additionally, we implemented a closed-loop control system to maintain the temporal stability of spectral measurements. This approach enabled the development of an innovative Brillouin

microscope capable of high spectral precision and long-term, stable, fully automated acquisitions, an outcome that was previously unattainable. We then conducted measurements on living cells exhibiting distinct types of biomolecular condensates across multiple experimental days: the EOM-BM successfully distinguished between physiological and pathological condensates based solely on their Brillouin frequency shift, outperforming standard techniques due to its label-free nature and enhanced spectral precision. To validate our Brillouin shifts measurements, we conducted parallel FRAP acquisitions: the correlation between methods revealed a liquid-to-gel phase transition of the condensates, consistent with a percolation model[4] and suggesting a fractal internal structure architecture. These results provide important advances into the physical properties of biomolecular condensates and establish a framework for their investigation with an enhanced Brillouin Microscope.

## Results

### EOM source implementation

The EOM is a fiber-coupled integrated photonic circuit that modulates the phase of the input laser source with an electric field generated by a microwave source, producing an output optical signal that can be frequency-shifted in the GHz range from the carrier. Figure 1C shows the layout of our standard BM (blue elements) in which we inserted the EOM launch, source modulation and collection (red elements), thus obtaining a Brillouin Microscope equipped with an EOM (EOM-BM). The frequency-modulated EOM output was coupled into the collection fiber and then delivered, together with the Brillouin signal coming from the sample, to the spectrum analyzer (based on a standard double-VIPA layout[23]).

The EOM signal did not interact with the sample and functioned solely as a frequency reference during Brillouin measurements. In our setup, its frequency ($v_{EOM}$) could be selected within the range of 0–15 GHz, providing high flexibility for different applications. Additionally, the EOM peaks could be used to estimate the spectrometer's resolution and line shape, allowing point-by-point deconvolution of the Brillouin spectra and enabling quantification of the sample's intrinsic Brillouin linewidth[44].

The right inset of Fig. 1C shows two representative Brillouin spectra of water acquired with the EOM-BM at different times. Here, the EOM frequency was set to 13 GHz to avoid overlap with the Brillouin signals in the frequency ranges relevant for biological materials (i.e., 7.50–9.50 GHz for tissues[29,30,32] and 7.50–8.10 GHz for cells[24,25,27–29,36] using a 532 nm laser). The spectra, recorded five min apart, exhibited a shift (~50 MHz) that could be quantified by tracking the EOM peaks and that was significantly higher than the required spectral precision (i.e., ≤10 MHz).

The instrument was governed by a custom-developed software that allowed for data acquisition and Brillouin maps reconstruction. In the EOM-BM this software contained an additional automatic routine that governed the EOM frequency, performed automatic pixel-to-GHz calibrations (described in the next section) and prevented spectral drifts in time. Specifically, this last point was achieved by combining a finely (~GHz) tunable laser source with a closed-loop feedback control[43] (herein called "EOM control loop") which dynamically adjusted the laser frequency. This procedure is sketched in the lower panel of Fig. 1C and is the core of the EOM-BM acquisition routine. Briefly, after a manual alignment and automatic calibration of the spectrometer, we started a Brillouin dataset with a specific $v_{EOM}$. At fixed intervals (either before starting a new acquisition or during a measurement) the EOM control loop monitored the position of the EOM Anti-Stokes peak: if its position shifted from the initial more than a threshold, the laser frequency was tuned until the EOM peak was restored to the original, thereby compensating for the drift; if no shift was detected, the laser frequency remained unchanged. In our protocol, this threshold was set at 50 MHz as an empirical compromise

between two competing requirements during data acquisition: ensuring sufficiently frequent recalibrations to maintain spectral accuracy and prevent Rayleigh line spillover, while minimizing acquisition interruptions and measurement time loss. The laser tunability, governed with our custom software, was thus used to re-center the Brillouin spectrum within the spectrometer detection window without requiring any manual opto-mechanical intervention. As detailed next, after every EOM control loop the system performed an automatic pixel-to-GHz recalibration to maintain measurement accuracy. As a result, acquisitions could autonomously proceed without spectral drifts, with the EOM control loop running at user-defined intervals (typically, every ~10'–30').

### EOM source as a sample-free calibrator

In the measurement pipeline, we exploited the known frequencies generated by the EOM as a strategy for precise pixel-to-GHz spectrometer calibration, independent of any sample. When the EOM is modulated at a single frequency $v_{EOM}$, sidebands at both the fundamental and its second harmonic are added to the input monochromatic light. In the EOM-BM system, equipped with a double-VIPA setup, a single modulation frequency therefore produces four peaks, as shown in Fig. 2A (top panel). This configuration enables the simultaneous extraction of both the VIPA dispersion curve and the FSR from a single EOM frequency.

In order to reconstruct the VIPA calibration curve with more accuracy, we modulated the EOM at 4 frequencies covering the whole range of interest for biological applications of Brillouin Microscopy, obtaining a total of 16 peaks (Fig. 2A, lower panel). The corresponding pixel-to-GHz calibration curve is shown in Fig. 2B, where we used the Anti-Stokes positions (circles in the graph) for the determination of the coefficients of the dispersion curve, and the remaining Stokes peaks (squares in the graph) for the quantification of the FSR. This procedure had great precision in determining the pixel-to-GHz relation, particularly in the frequencies of interest for Brillouin Microscopy (lower inserts of Fig. 2B). In Fig. 2C we compared the calibration curve obtained with the EOM (red marks and curves) with the curve obtained from the standard calibration protocol with water and methanol Brillouin shifts[20,23] (blue marks and curves, detailed in Supplementary Fig. 1C). With the EOM, we achieved ~10 times higher precision in determining the Brillouin shifts dispersion across the entire FSR.

This calibration protocol, moreover, was very rapid (less than 5 seconds were needed to obtain all the shown curves) and could be programmed in the data acquisition routine in order to automatically calibrate the system at any desired time. If more precision is required, additional EOM frequencies can be added, ensuring high flexibility for different scopes.

In summary, the calibration protocol of the EOM-BM system presented here is entirely independent of reference samples, which are known to be affected by ambient temperature fluctuations[38] and by the microscope objective used[39]. Instead, its pixel-to-GHz calibration curve is based on absolute frequency references and thus reflects only the spectrometer alignment. This method is highly precise, rapid, and fully automated. It can be applied to calibrate the pixel-to-GHz dispersion curves of spectrometers regardless of their specific configurations, making it compatible with single-VIPA, double-VIPA, and tandem Fabry–Perot setups[37]. This innovative approach offers a standardized calibration framework that may enhance the reproducibility of Brillouin measurements across different laboratories, an essential requirement for advanced biological applications of Brillouin microscopy[20,23].

### Temporal stability of the EOM-BM

In this section, we demonstrate that the method introduced, based on a closed-loop feedback control (i.e., the "EOM control loop" of

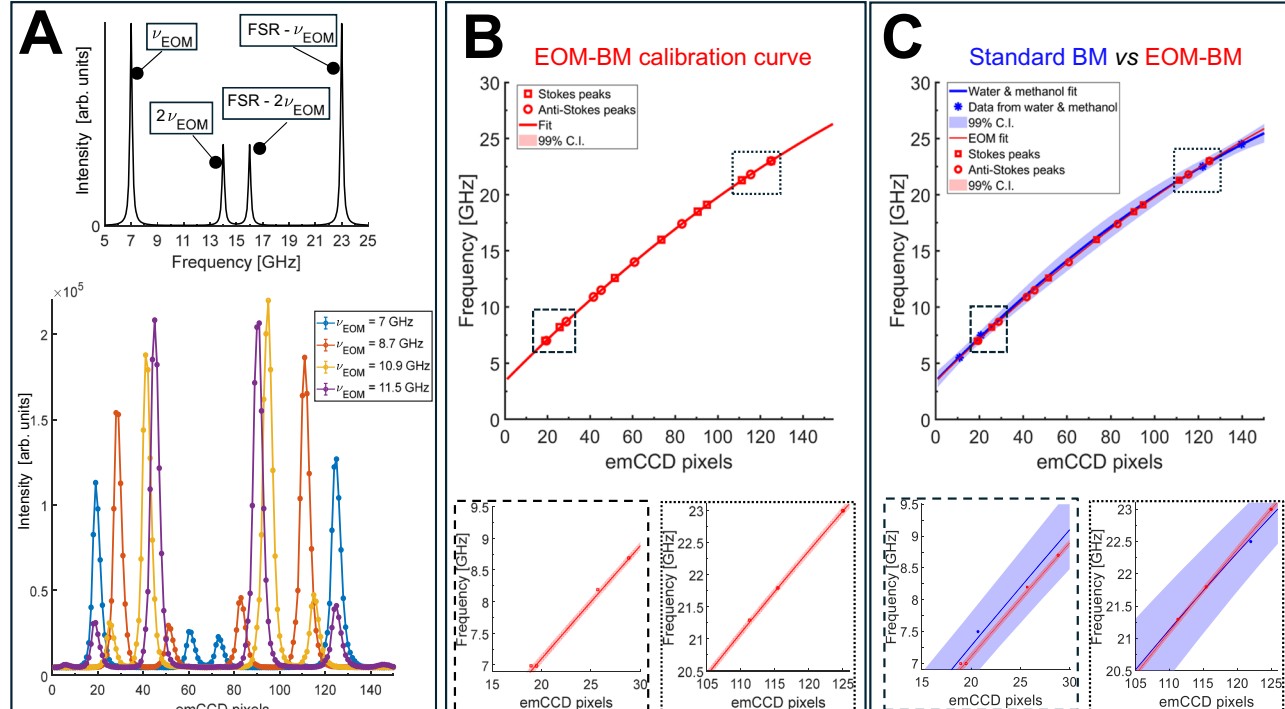

**Fig. 2 | The EOM source can be used as a reference-free pixel-to-GHz calibrator of Brillouin spectrometers. A** upper panel: schematic output of a single EOM frequency (here $v_{EOM} = 7$ GHz) with a 532 nm laser: it was constituted by 4 peaks, i.e., the fundamental and its second harmonics (i.e., $v_{EOM} = 7$ GHz and $2*v_{EOM} = 14$ GHz) and their respective Stokes peaks (i.e., FSR-$v_{EOM} = 23$ GHz and FSR-$2*v_{EOM} = 16$ GHz, with FSR = 30 GHz). Lower panel: EOM spectra used for pixel-to-GHz calibration of our EOM-BM, coming from $v_{EOM} = 7$ GHz (blue spectrum), $v_{EOM} = 8.7$ GHz (red), $v_{EOM} = 10.9$ GHz (yellow) and $v_{EOM} = 11.5$ GHz (purple). **B** from the spectra of panel A, we had a total of 16 datapoints: 8 of them (Stokes peaks, located at $v_{EOM}$ and $2*v_{EOM}$, circle data points) were used to retrieve the pixel-to-GHz dispersion curve, while the remaining (Anti-Stokes peaks, located at FSR-$v_{EOM}$ and FSR-$2*v_{EOM}$, square datapoints) were used to retrieve the FSR. The first and last pairs of datapoints overlapped, as they corresponded to Stokes and Anti-Stokes signals arising from different EOM fundamental frequencies. The continuous line is the 2nd order polynomial fit, shaded areas show 99% confidence intervals of the curve. Inserts show the fit accuracy in the frequencies of interest for biological applications (both Stokes and Anti-Stokes sides). **C** comparison of the dispersion curves obtained with our EOM-BM (red datapoints, line and shaded areas) with the standard protocol from water and methanol Brillouin shifts (blue datapoints, line and shaded areas, see Supplementary Fig. 1C). The continuous lines are 2nd order polynomial fits, shaded areas show 99% confidence intervals of this curve: the errors associated with our EOM-BM are ~10 times lower than those of the standard protocol across the entire frequency range of a BM, and especially in the frequencies of interest for biological applications (at 7.50 GHz, $C.I._{EOM} = 60$ MHz vs $C.I._{water\&methanol} = 540$ MHz). Source data are provided as Source Data file.

Fig. 1C), effectively eliminates spectral drifts. This is achieved by monitoring the position of the EOM Anti-Stokes peak at desired intervals over time and dynamically adjusting the laser frequency as necessary.

Figure 3A (first panel) shows a representative acquisition performed without the EOM control loop and without recalibration, simulating the behavior of a standard Brillouin microscope. In this case, data were acquired from a water sample maintained at a constant temperature ( ± 0.2 °C) over time. Despite the stable sample conditions, significant spectral drifts were observed as early as 100 min after the initial spectrometer alignment and calibration, resulting in Rayleigh line spillover (yellow spectra). This degradation forced an interruption of the acquisition and required manual realignment of the spectrometer to proceed with the measurement.

These spectral drifts were primarily caused by small fluctuations in room temperature (Fig. 3A, second panel), which affected the mechanical stability of the spectrometer. If left uncorrected, this led to the intrusion of a substantial elastic tail background into the Brillouin channel, ultimately distorting the spectra and compromising the accuracy of both $v_B$ and $\Gamma_B$ measurements (Fig. 3A, third panel). The EOM reference peak exhibited a large shift from its initial position, approximately 12% over 100 minutes (Fig. 3A, fourth panel), while FSR, calculated from the separation between the EOM peaks, showed a clear dependence on ambient temperature (Fig. 3A, fifth panel). This behavior, also observed with a frequency-locked laser (see Supplementary

Fig. 1B), indicates that the initial pixel-to-GHz calibration became invalid over time.

To evaluate the temporal stability of the EOM-BM system, we conducted a 50-h continuous acquisition in which Brillouin spectra of water were recorded every 15 minutes. For each acquisition, both the EOM control loop and spectrometer calibration were executed at the start. Representative spectra from the first, 24 h, and 48 h timepoints are shown in Fig. 3B (first panel). During the course of the measurement, the room temperature varied, while the water sample was maintained at a constant temperature (Fig. 3B, second panel). Under these conditions, the Brillouin shift and linewidth (Fig. 3B, third panel), as well as the EOM reference position (Fig. 3B, fourth panel), exhibited significantly improved stability compared to the standard BM setup. Specifically, the EOM peak position shifted by less than 0.4%, approximately 30 times smaller than the drift observed without EOM control loop.

The system's automated recalibration preserved a consistent FSR throughout the entire acquisition, effectively decoupling the calibration from ambient temperature fluctuations (Fig. 3B, fifth panel). The insets highlight the performance of the EOM-BM over the same time-scale as in Fig. 3A, clearly demonstrating its superior stability compared to conventional Brillouin microscopes.

Thus, the EOM-BM enabled automated, periodic checks and corrections of the laser and spectrometer status: from a standard BM stable for less than 30 min (i.e., the time at which $v_B$ changed more than

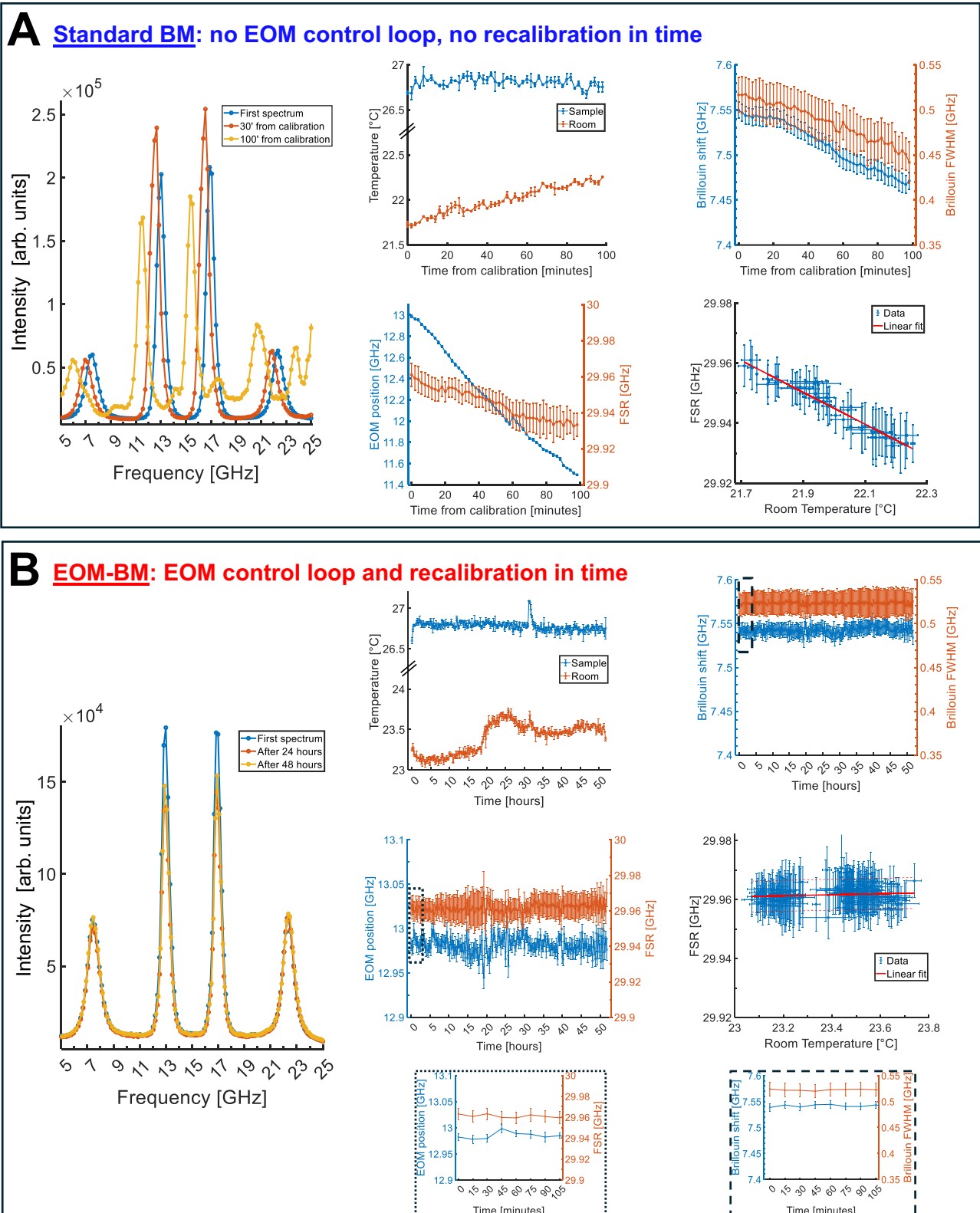

0.1% from its initial value, Fig. 3A), we obtained an instrument stable for more than 2 days (Fig. 3B, where the acquisition was stopped only due to time constraints).

Importantly, the EOM control loop alone allowed only to restore the initial state in which Brillouin peaks had equal intensity, thus continuing the acquisitions without drifts and preventing camera damage. In cases where a tunable laser cannot be used, the same outcome might be achieved by inserting a motor to adjust the VIPA angle and implementing the feedback control loop on this parameter, rendering this method universal for any BM.

Instead, frequent pixel-to-GHz calibrations were essential to ensure the validity of the calibration parameters during time and thus a reliable data analysis. In fact, when performing only the EOM control loop without the spectrometer calibration at the beginning of every

**Fig. 3 | Our EOM-BM shows superior spectral stability over time compared to a standard BM. A** first panel: Brillouin spectra of water (kept at constant temperature with an incubator) acquired at different times from spectrometer alignment and calibration. Here, no EOM control loop was implemented, simulating the behavior of a standard BM. Eventually, unfiltered Rayleigh appeared at the edge of the camera and resulted in signal distortion: after 100′, the acquisition had to be stopped and the spectrometer manually realigned. Second panel: temperature changes in sample (blue) and room (orange, with <1 °C drift in 100′). Third panel: water Brillouin shift (blue) and FWHM (orange) in time. Because of temporal instabilities, $v_B$ and $\Gamma_B$ considerably drifted ($v_B$ changed ~1%, much more than the requested 0.1%). Fourth panel: EOM Anti-Stokes peak position (blue) drifted ~1.5 GHz over 100′ (~12% change). FSR (orange) showed only minor variations (~30 MHz, i.e., ~0.1% change); nevertheless, this was enough to cause time-dependent changes of Brillouin parameters. Fifth panel: FSR showed strong dependence on room temperature (correlation coefficient = 0.99); this effect was also observed with a frequency-locked laser (Supplementary Fig. 1B). **B** first panel: Brillouin spectra of water (kept at constant temperature with an incubator) acquired at different times with our EOM-BM protocol, performing EOM-control loop and calibration at the beginning of every acquisition (as sketched in Fig. 1C). Here, Brillouin spectra remained stable for more than 2 days. Second panel: temperature changes of sample (blue) and room (orange) in time. Third panel: Brillouin shift (blue) and FWHM (orange) in time, showing superior stability than panel A. Fourth panel: EOM position (blue) shifted <50 MHz, i.e., ~0.4%; thanks to repeated calibrations, FSR (orange) remained constant, showing only ~1*10^{-5}% change. Fifth panel: FSR did not show any dependency from room temperature. Squared inserts: EOM position, FSR, Brillouin shift and FWHM in the first 100′, the same timescale as panel A. All data are shown as mean ± SD performed over 500 repeated measurements of a single acquisition; time = 0 refers to the spectrometer manual alignment and calibration (panel A) or just manual alignment (panel B). Source data are provided as Source Data file.

acquisition (as shown in Supplementary Fig. 3), periodic oscillations of Brillouin shifts and widths, not related to sample properties, could be observed over time; moreover, FSR showed the same dependency from room temperature previously seen (Fig. 3A, and Supplementary Fig. 1B). As a result, in our routine any feedback action was always followed by a calibration.

Taken together, the stability and ease of use of our EOM-BM represents a step forward in standard BMs that allow automatic data acquisitions for extended periods of time. This technological improvement allowed to have stable and reliable Brillouin values of water or maps of cells, as illustrated in the next sections.

## Temperature dependence of the water sound velocity and damping

To assess the reliability of the EOM-BM system in detecting subtle changes in sample properties, we measured the temperature dependence of the Brillouin shift ($v_B$) and FWHM ($\Gamma_B$) of water (Fig. 4). Notably, this acquisition was fully automated and spanned approximately 12 h, during which no manual adjustments to the instrument were required. From accurate spectral fits at each temperature point (two representative spectra of the acquisition are shown in Fig. 4A), we extracted $v_B$ and $\Gamma_B$ (Fig. 4B), which were subsequently used to calculate[45] water speed of sound ($V_L$, Fig. 4D) and longitudinal kinematic viscosity ($v_L$, Fig. 4E). During the 12 h measurement, the EOM position and FSR remained highly stable (Fig. 4C).

The temperature dependence of the speed of sound in water ($V_L$) is known with high precision and has been used as a benchmark to evaluate the accuracy of custom-built Brillouin microscopes globally[20]. Our $V_L$ measurements closely followed the theoretical model curve[38] and data acquired with acoustic spectroscopy[20,46], demonstrating higher accuracy than many previously reported systems[20].

For the longitudinal kinematic viscosity ($v_L$), its theoretical temperature dependence is challenging to model accurately. Therefore, we compared our Brillouin-derived values with data obtained from ultrasound measurements[47,48], Inelastic X-ray Scattering[47,49], and estimates derived from other experimental datasets using various approximations[47,50]. All experimental datasets fell within the 99% confidence interval of the values measured using our EOM-BM system. Importantly, $\Gamma_B$ used for this estimation corresponded to the deconvolved FWHM, obtained by using the EOM line shape as estimation of the spectrometer point spread function[44]. The agreement with other results is thus particularly noteworthy, as the Brillouin FWHM is highly sensitive to both data quality and the accuracy of the fitting model. Deviations from expected values are common in other spectrometers, where larger associated error bars have been reported[20].

In summary, these tests confirm the reliability and stability of the EOM-BM in detecting subtle temperature-induced changes in samples. Moreover, they highlight the limitations of using water for spectrometer calibration, since its Brillouin shift is highly sensitive to temperature variations, potentially causing observed day-to-day fluctuations in the calibration curve on the order of several MHz.

## High-resolution Brillouin imaging of ALS/FTD-related biomolecular condensates in living cells

We finally applied the EOM-BM system to investigate changes in the Brillouin shifts of biomolecular condensates implicated in amyotrophic lateral sclerosis (ALS) and frontotemporal dementia (FTD)[2,4,51]. To this end, we expressed relevant proteins in SK-N-BE cells, a neuroblastoma cell line commonly used as a model for neurodegenerative diseases, and performed Brillouin microscopy on living cells. To validate the mechanical information obtained from the Brillouin maps, we conducted parallel Fluorescence Recovery After Photobleaching (FRAP) experiments. FRAP is a well-established technique for assessing protein mobility and dynamics; the time-dependent recovery of fluorescence intensity is correlated with changes in mechanical properties, ranging from liquid-like behavior, characterized by rapid and/or complete recovery, to solid-like states under pathological conditions, which exhibit slower and/or incomplete recovery[5–8,11,12,16,52–58].

The investigation using BM and FRAP focused on two proteins with distinct structural and biological features but a shared propensity to form insoluble and toxic aggregates in response to mutations or post-translational modifications[2,4]. The first was superoxide dismutase 1 (SOD1), a protein whose p.A4V pathogenic variant is associated with one of the most aggressive familial forms of ALS[53,59–61]. The second was TAR DNA-binding protein (TDP-43), whose C-terminal fragments encompassing aminoacidic residues 209–414 or 220–414 (herein called TDP-43[209] and TDP-43[220], respectively) are detected in postmortem brain tissues of individuals affected by ALS or FTD[62–66]; in particular, TDP-43[220] is among the most commonly detected and characterized fragments[62,63,65–68]. Both TDP-43[209] and TDP-43[220] were individually expressed upon stable genomic integration of the corresponding coding vectors and thanks to a doxycycline-inducible promoter, fused to a GFP to enable their localization via fluorescence microscopy: thus, the GFP signal was highly specific for the protein of interest. As a control, we included Ras-GTPase-activating proteinbinding protein1 (G3BP1), a protein marker of SGs core, which is essential for the LLPS-mediated assembly and dynamics of physiological SGs[3,11,12,14]. GFP alone, expressed by the same vector, served as an additional negative control to ensure that GFP itself neither underwent phase separation under stress, nor affected cellular mechanical properties. To mimic the oxidative stress known to trigger aggregates formation in ALS/FTD[69], cells were treated with sodium arsenite (ARS)[58,70]. All proteins analyzed with BM and FRAP, along with their corresponding phase transitions, are shown in Fig. 5A.

Our custom-built EOM-BM, equipped with a top stage incubator for temperature, humidity and $CO_2$ control, enabled biomechanical imaging of living cells in a physiological environment (shown in Fig. 5B; more Brillouin maps can be seen in Supplementary Fig. 4). Alongside

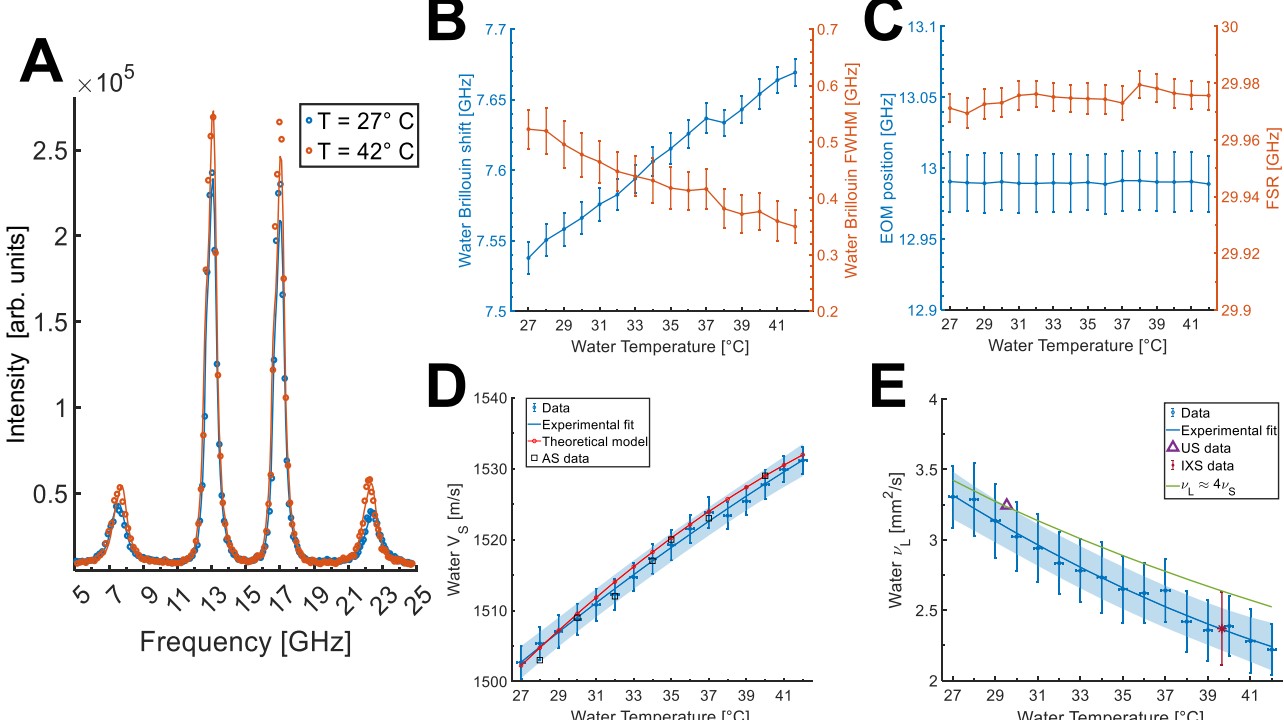

**Fig. 4 | reconstruction of water Brillouin dependency with temperature.** The same measurement has been used as a benchmark for evaluating the reliability of standard BMs from all over the world[20]. Importantly, this acquisition was completely automatic and lasted ≈12 h, during which the spectrometer was never realigned. After the temperature setpoint was reached, the system automatically performed the EOM-control loop and calibration and finally acquired 500 Brillouin spectra, then waited for the next setpoint in a recursive manner. **A** example of 2 Brillouin raw spectra (dots) extracted from the data at 27 °C (blue data) and 42 °C (red data) together with corresponding fits (continuous lines). The EOM frequency was set to $\nu_{EOM} = 13$ GHz. **B** Water Brillouin shift (blue) and deconvolved FWHM (orange) dependency with temperature. **C** EOM Anti-Stokes peak position (blue) and FSR (orange) in function of temperature, both remaining stable throughout 12 h of experiment. **D** water velocity of sound ($V_S$) extrapolated from Brillouin shifts

(blue datapoints and curve); in red, theoretical model[38]; black squares show acoustic spectroscopy (AS) data[20,46]. Our data match the model curve and AS data within 99% confidence intervals (shaded blue areas). **E** Water longitudinal kinematic viscosity ($\nu_L$) extrapolated from deconvolved Brillouin FWHMs (blue datapoints and curve). We compared our results with already published data from ultrasounds (US)[47,48] (purple triangle) and Inelastic X-ray Scattering (IXS)[47,49] (red star). We also show the curve of the approximation $\nu_L \sim 4\,\nu_S$ (where $\nu_s$ are water kinematic shear viscosity data[47,50]). Our data match both US and IXS data within 99% confidence intervals (shaded blue areas) and are slightly lower than $\nu_L \sim 4\,\nu_S$ approximation curve. Data of panels B-E are shown as mean ± SD performed over 500 repeated measurements of a single acquisition; shaded areas refer to 99% confidence intervals of the fits. Source data are provided as Source Data file.

BM data, we simultaneously acquired brightfield and fluorescence images at the same focal plane, facilitating accurate localization of condensates. Using fluorescence images in the post-processing analysis, we applied binary masks (Supplementary Fig. 5) allowing for precise quantification of the Brillouin shift of the condensates (whose distributions are shown in Fig. 5B, lower panels). Collectively, our BM data showed abnormal Brillouin shifts of SOD1[A4V] and TDP-43 C-terminal fragments in stress conditions (commented in detail in Supplementary Note 1), consistent with extensive literature regarding their insolubility and toxicity in cells[53,61,63–68,71–77]. The technological enhancements provided by the EOM-BM allowed for automatic BM acquisitions on living cells and consistent quantification over extended experimental sessions, allowing reliable comparisons across multiple days and biological replicates.

FRAP data (Fig. 5C, and Supplementary Fig. 6, Supplementary Fig. 7) were obtained on the same batch of cells by photobleaching a specific region within the GFP-labeled condensates and monitoring its intensity recovery over time. Recovery curves (shown in Supplementary Fig. 7A) of every condensate were fit to extract two parameters: the immobile fraction (Fig. 5C, lower panels) and the half-recovery time (Supplementary Fig. 7B)[21,78,79]. Throughout the experiments, the size of FRAP ROIs was kept constant to ensure comparability across acquisitions. As shown in Supplementary Fig. 7C, we verified the robustness of our data by comparing recovery curves from

condensates of varying sizes. Even when their dimensions were smaller or larger than the FRAP ROIs, the overall recovery dynamics and parameters remained consistent, thus suggesting that variations in condensates size did not bias our analysis. Results of FRAP data are reported in detail in Supplementary Note 1.

Interestingly, we observed a strong correlation between the Brillouin shifts and the FRAP immobile fractions of the various condensates: higher Brillouin shifts, indicative of increased longitudinal elastic modulus, were associated with higher FRAP immobile fractions, reflecting reduced molecular mobility within the condensates. This relationship suggests that as the mechanical rigidity of SGs increases, their internal molecular dynamics become progressively constrained. Figure 6 presents a plot of Brillouin shifts against immobile fractions, revealing a power-law dependence between the two parameters (mathematical details are given in Supplementary Note 1) typical of a percolation phenomenon[9,13,80]. This behavior aligns with theoretical models describing SGs phase transition from a liquid-like to a gel-like state as a combination of LLPS and networking transitions such as percolation and gelation[4,13,80–82] through the formation of a cross-linked molecular network[3,4,14,83–85]. This networking transition is characterized by a concentration threshold, known as the percolation threshold, that defines the gel point[4,14,80,85]: from a fit of our data (Fig. 6) we extrapolated both the percolation threshold of this process and the power-law exponential β, whose fitted value is coherent with

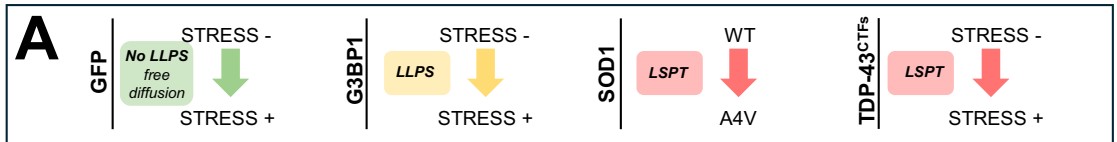

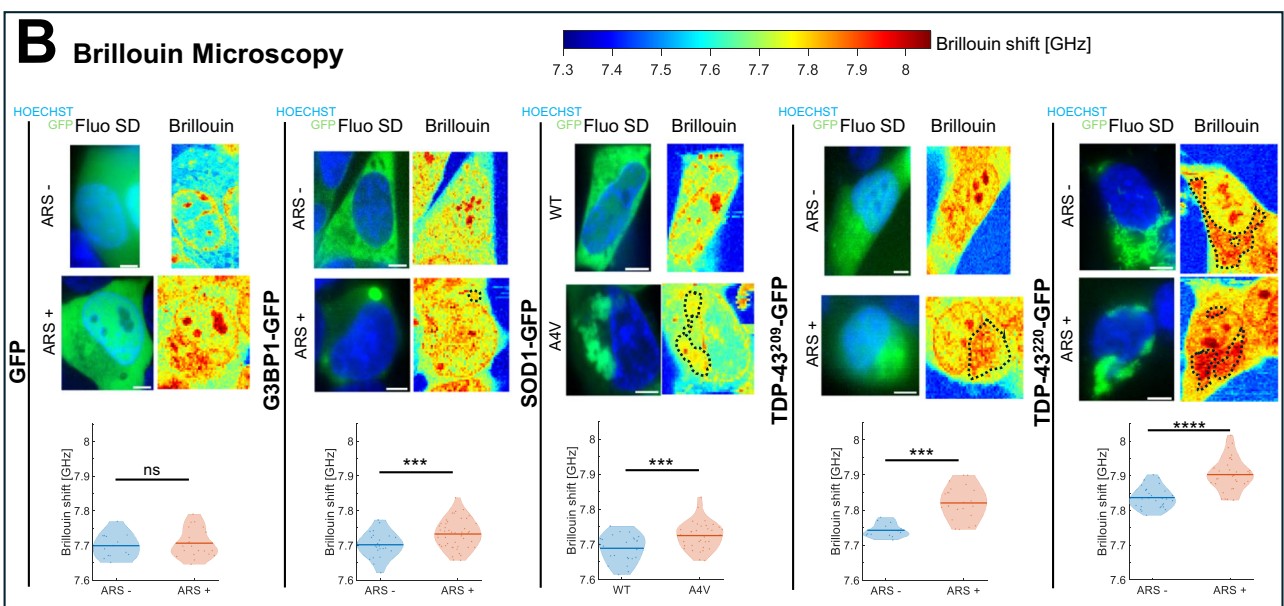

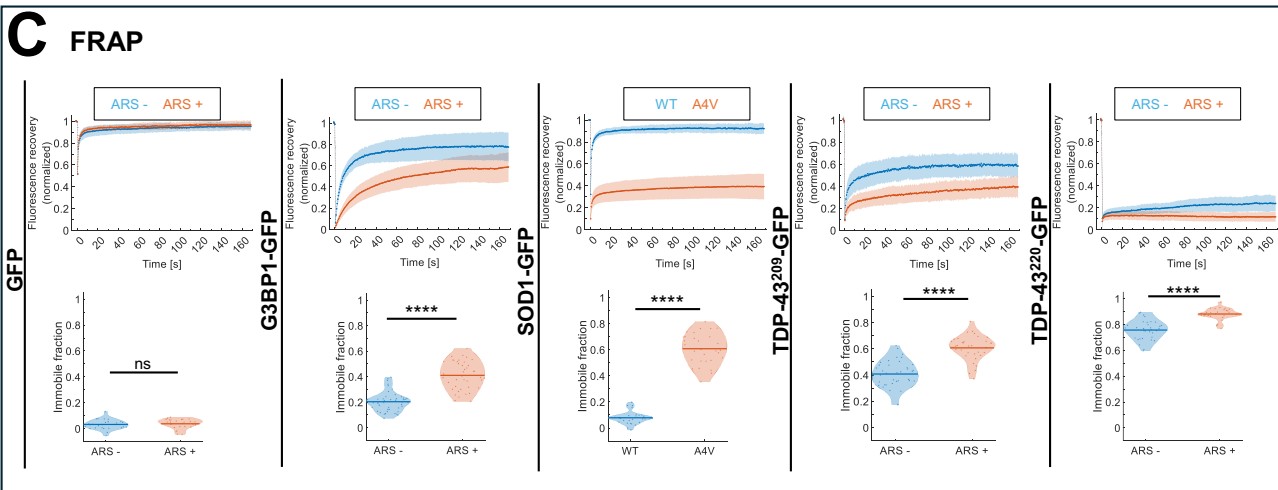

**Fig. 5 | Brillouin Microscopy and FRAP data acquisitions on living SK-N-BE cells expressing different physiological and pathological biomolecular condensates. A**: summary of the different exogenous proteins expressed in the cells, together with their phase transitions. LLPS: liquid-liquid phase transition; LSPT: liquid-solid phase transition; CTF: C-terminal Fragments. **B**: Brillouin Microscopy data acquired on living cells with our EOM-BM setup. Upper panels: representative Brillouin shift maps of cells expressing the different proteins, in absence or presence of stress (i.e., sodium arsenite, ARS) or with a malignant mutation (A4V); additional maps are shown in Supplementary Fig. 5. Together with Brillouin shift maps, Fluorescence images of nuclei (live-stained with HOECHST) and proteins labeled with GFP are also shown. Dashed black contours highlight granules location in Brillouin maps. Scale bars = 5 μm. Lower panels: violin plots summarizing Brillouin shifts distributions of the condensates in each condition, including statistical analysis. Data were obtained from n ≥ 3 independent biological replicates (in every replicate, at least 5 cells were acquired); solid lines show the mean of the distributions. GFP: $p = 0.7$, Wilcoxon rank-sum test; G3BP1-GFP: $p = 0.0027$, t-test;

SOD1-GFP: $p = 0.003$, Wilcoxon rank-sum test; TDP-43[209]-GFP: $p = 2.5*10^{-6}$, t-test; TDP-43[220]-GFP: $p = 8*10^{-9}$, t-test. All tests were two-sided. These shifts were obtained for each cell by averaging over a binary mask derived from GFP fluorescence signal (see Supplementary Fig. 6). **C** FRAP curves acquired on a single region of fluorescently-labeled condensates in living cells. Upper panels: FRAP data are shown as mean (solid curve) ± SD (shaded areas), obtained from $n = 3$ independent replicates (in every replicate, at least 10 cells were acquired); see Supplementary Fig. 7 for examples of FRAP acquisition. Lower panels: violin plots of immobile fraction distributions of the data in each condition, obtained from the fit of single FRAP curves of every cell (Supplementary Fig. 7A), together with statistical analysis. GFP: $p = 0.32$, Wilcoxon rank-sum test; G3BP1-GFP: $p = 1*10^{-12}$, Wilcoxon rank-sum test; SOD1-GFP: $p = 1*10^{-12}$, Wilcoxon rank-sum test; TDP-43[209]-GFP: $p = 4*10^{-13}$ t-test; TDP-43[220]-GFP: $p = 7*10^{-9}$ Wilcoxon rank-sum test. ns not significative; ***: $p < 0.005$; ****: $p < 0.0001$. All tests were two-sided. Source data are provided as Source Data file.

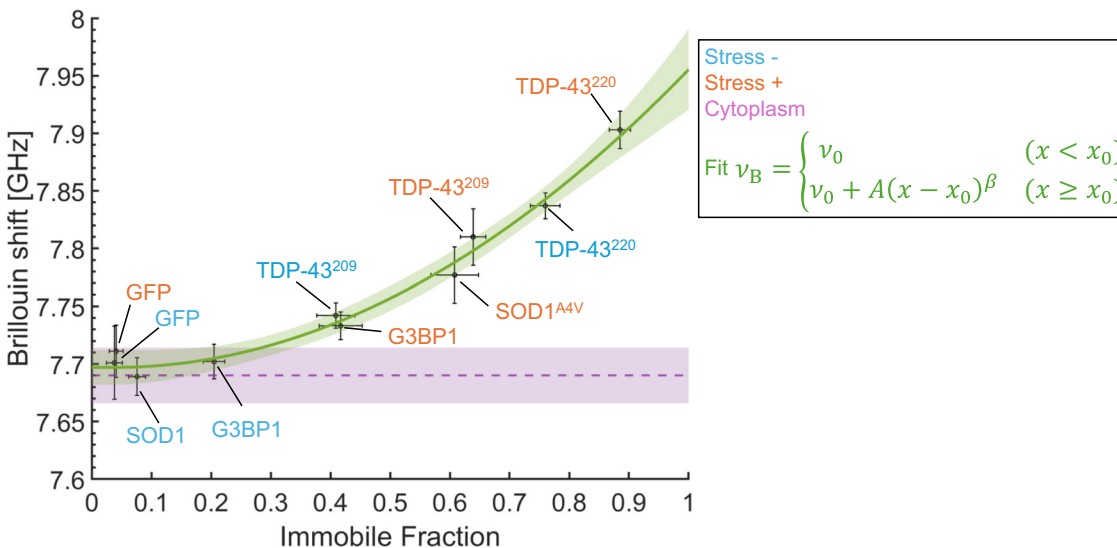

**Fig. 6 | Correlation plot of Brillouin shifts versus FRAP immobile fractions reveals a percolation-like behavior and a fractal internal structure of protein condensates in living cells.** The mechanical properties of liquid-gel phase transition of biomolecular condensates such as stress granules have been explained as a combination of LLPS and networking transitions such as percolation and gelation[4,13,80,81]. Data were fitted with a power-law model characteristic of a percolation transition, allowing us to extract the percolation threshold $x_0$ (defining the gel point; from the fit, $x_0 = 0.04 \pm 0.2$) and the exponent $\beta$ (here, $\beta = 2.0 \pm 0.2$, consistent with a 3D gelling system undergoing percolation[86]). See Supplementary

Note 1 for further details. Violet plots: mean (dashed line) ± 2*S.E.M. (shaded area) Brillouin shifts of cytoplasm of control SK-N-BE cells not overexpressing proteins (shown in Supplementary Fig. 4). Green plots: power-law fit (solid line), with 95% C.I. (shaded area). Data labels are colored blue (condensates without stress) or orange (condensates under stress, e.g., ARS treatment or A4V mutation). Data points are shown as mean ± 2*S.E.M, obtained from $n = 3$ biological independent replicates (in every replicate, at least 10 cells for FRAP or 5 cells for Brillouin were acquired). Source data are provided as Source Data file.

theoretical models describing the elastic properties of gelling systems[86]. In this framework, SGs mechanical and dynamic properties are governed not solely by molecular composition, but also by the emergent architecture of molecular interactions. Our data thus support several models in which the elastic properties of SGs are dictated by an underlying fractal network structure[15,83], whose spatial organization implies that SGs are not homogeneous droplets, but rather possess a hierarchical internal architecture, where mechanical stiffness and molecular diffusivity are scale-dependent[83]. Importantly, these models have lacked supporting experimental data in physiological cellular environments; our findings thus represent a significant experimental evidence of the fractal internal architecture of protein aggregates in living cells, shedding new light on their physical properties.

The EOM-BM represented a unique method for exploring the fractal properties of protein condensates: as detailed in Supplementary Note 2, it outperformed standard methods for characterizing condensates mechanics. Indeed, we validated Brillouin measurements of Fig. 5B on the same cells after fixation (shown in Supplementary Fig. 8), demonstrating the applicability of EOM-BM on fixed samples. This represents a key advantage over FRAP, limited to live cells with actively mobile proteins[79].

Furthermore, in an additional BM-FRAP dataset on the same G3BP1-GFP cells with inducible co-expression of pathogenic FUS[P525L] protein (associated with severe ALS[16]), Brillouin Microscopy revealed intrinsic mechanical features inaccessible to FRAP (Fig. 7; Supplementary Note 2). The presence of FUS[P525L] within G3BP1-GFP granules was confirmed via immunofluorescence (shown in Supplementary Fig. 9 [87,88]). Brillouin shifts successfully distinguished physiological, liquid-like G3PB1-GFP condensates from their pathological, gel-like counterpart formed in presence of FUS[P525L]; instead, FRAP curves showed no statistically significant differences between the two cases. These data highlight the unique capabilities of Brillouin Microscopy, particularly its label-free nature, and underscore its broader potential in detecting protein phase separation: in particular, this technique

enables to assess the properties of the entire condensate compartment containing multiple protein species, rather than being limited to the tagged protein of interest as in fluorescence-based approaches. FUS[P525L] may therefore be crucial in determining the mechanical properties of G3BP1 aggregates, readily detected by our EOM-BM but indistinguishable using FRAP.

With the EOM-BM, we thus moved from Brillouin maps of cells affected by severe drifts (as in Fig. 1B or Supplementary Fig. 1D), which hindered the extraction of meaningful biomechanical information, to precise and accurate mechanical maps of various condensates in living and fixed cells. Taken together, these results support the hypothesis that a liquid-to-gel phase transition contributes to the formation of aberrant biomolecular condensates[2,4], providing an important experimental evidence of the fractal internal architecture of protein aggregates in living cells. This underscores the potential of Brillouin Microscopy, when implemented with the necessary spectral precision and stability, as a powerful and unique tool for investigating protein phase separation, surpassing the capabilities of conventional techniques.

## Discussion

Stress granules (SGs) are micron-sized biomolecular condensates that physiologically form in the cytoplasm as a response to cellular stress and spontaneously disassemble upon stress resolution. Alterations in their mechanical properties can trigger a phase transition from a liquid-like to a gel-like state: the former is reversible and physiological, while the latter is irreversible and associated with severe neurodegenerative diseases (such as ALS and FTD[2,4]). Despite their biomedical relevance, SGs mechanical characterization is limited by the lack of techniques capable of probing sub-cellular compartments mechanics with adequate resolution[2,5,15,17]. Indeed, established approaches used to quantify biomechanical properties (e.g., passive micro rheology, coalescence observations, Atomic Force Microscopy (AFM), and optical tweezers) depend on direct mechanical interaction with the specimen[5,17–20]. AFM is considered

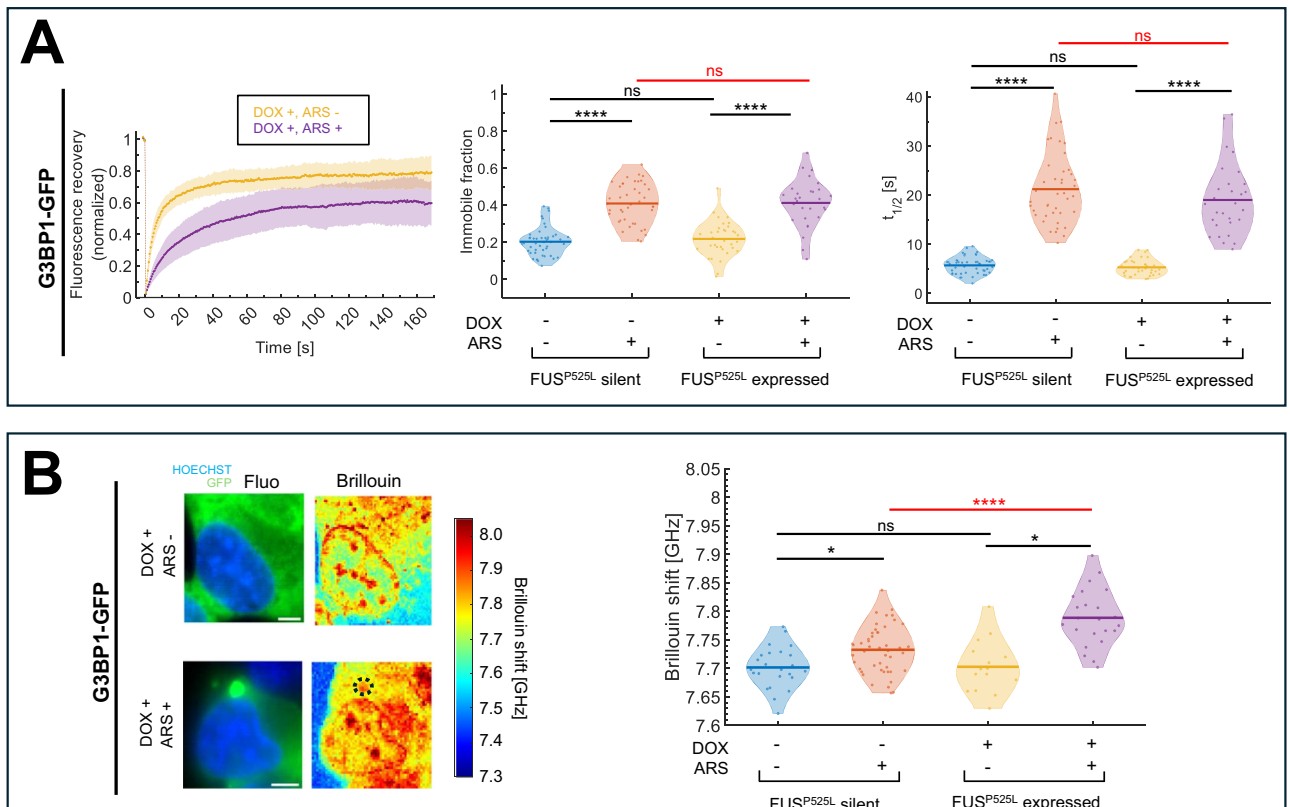

**Fig. 7 | Brillouin Microscopy and FRAP data acquisitions on G3BP1-GFP condensates in presence or absence of doxycycline-induced aberrant FUSP525L.** Here, FUSP525L presence in G3BP1-GFP granules was confirmed via immunofluorescence (Supplementary Fig. 9; Supplementary Note 2). **A** First panel: FRAP data are shown as mean (solid curve) ± SD (shaded areas), obtained from $n = 3$ independent biological replicates (in every replicate, at least 10 cells were acquired). Second panel: immobile fractions of G3BP1-GFP condensates without doxycycline (blue and orange distributions, also shown in Fig. 5C, second panel) and with doxycycline (yellow and violet distributions). Here, ARS treatment increased immobile fractions in both DOX- ($p = 5*10^{-12}$) and DOX+ ($p = 1*10^{-5}$) samples; instead, DOX treatment (inducing FUSP525L expression) did not modify the mobility of either ARS- ($p = 0.94$) or ARS+ ($p = 0.998$, red bar and text) samples. Statistical analyses were performed with two-sided Kruskal-Wallis tests followed by multiple comparison post-hoc tests. Third panel: half-recovery times of G3BP1-GFP condensates in DOX- (these data are shown in Supplementary Fig. 8A, second panel) and DOX+ cells. Likewise immobile fractions, ARS treatment increased half-

times in both DOX- ($p = 2*10^{-16}$) and DOX+ ($p = 1*10^{-20}$) samples, while DOX treatment did not modify times of either ARS- ($p = 0.998$) or ARS+ ($p = 0.997$, red bar and text) samples. Statistical analyses were performed with two-sided, one-way ANOVA followed by multiple comparison post-hoc tests. **B** Left: Brillouin shift maps of DOX + G3BP1-GFP cells untreated and treated with ARS, together with fluorescence maps of HOECHST (labeling nuclei) and GFP (labeling exclusively G3BP1). Black dotted shapes indicate granules in Brillouin maps. Scale bars = 5 μm. Right: Brillouin shifts of G3BP1-GFP condensates in absence (already shown in Fig. 5B, second panel) or in presence of FUSP525L, from $n = 3$ independent biological replicates (in every replicate, at least 5 cells were acquired). As in FRAP, ARS treatment increased Brillouin shifts of both DOX- ($p = 0.027$) and DOX+ ($p = 0.017$) samples. Unlike FRAP, however, DOX treatment increased Brillouin shifts of ARS+ condensates ($p = 2*10^{-5}$, red bar and asterisks); ARS- samples showed no differences ($p = 0.71$). Statistical analyses were performed with two-sided, one-way ANOVA followed by multiple comparison post-hoc test. ns not significant; *: $p < 0.05$; ****: $p < 0.0001$. Source data are provided as Source Data file.

the main technique for measuring the Young's Modulus $E$, whereby the material experiences a change in volume, through nano-indentation of the sample's surface. While it has been applied to biomolecular condensates in vitro[89,90], AFM lacks 3D capacities as its measurements are averaged along the axial direction and strongly depend on mathematical models to extract $E$[91]. Other techniques capable of in vivo capacities, such as Mechanosensitive fluorescent probes[92], report fluorescence changes in response to alterations in plasma membrane tension caused by external forces: consequently, they do not directly reflect the intrinsic mechanical properties of cellular structures. Similarly, tunable fluorophores like BODIPY-based molecular rotors are probes whose emission correlates with local viscosity, though it is also affected by solvent polarity, molecular aggregation or interactions with proteins and lipids[93,94]. Importantly, these probes are not label-free and require chemical incorporation into cells; moreover, their staining is confined to specific organelles or membrane regions, restricting their use in global or long-term mechanical measurements.

These limitations have motivated the development of high-resolution, non-invasive, label-free approaches for probing SGs mechanics in living cells. In this regard, Brillouin microscopy represents a promising candidate; however, its broader application is often limited by temporal instabilities, requiring constant manual supervision.

In this study, we developed and implemented an enhanced Brillouin Microscope (BM), termed EOM-BM, for high-resolution, long-term, and automated imaging of biomolecular condensates in living cells. By integrating an Electro-Optic Modulator (EOM) into a standard BM setup, we introduced three key functionalities: *(i)* stable frequency reference during spectral acquisitions, *(ii)* precise and automatic spectrometer calibration without reliance on unreliable standard samples, and *(iii)* real-time correction of temporal drifts through closed-loop feedback control.

Although a filtered Rayleigh signal might, in principle, provide a reliable frequency reference, implementations based on free-space interferometric filters (such as Michelson interferometers[95,96]) are

highly sensitive to mechanical vibrations and temperature fluctuations: thus, these systems require frequent (i.e., every 5-10 minutes) manual adjustments to maintain effective suppression in turbid media such as living cells. In contrast, EOM-generated peaks remain stable once aligned at the optical input and exhibit no temporal intensity fluctuations. This advancement enabled a shift from manually calibrated standard BM, requiring continuous user intervention and limited to short-term acquisitions, to a fully automated workflow capable of long-duration, stable measurements with minimal user input.

Notably, our EOM-based approach is compatible with various BM configurations[37], regardless of laser wavelength, tunability or spectrometer design, thereby offering great flexibility for diverse applications. A recent consensus paper[20] on Brillouin Microscopy for biological materials highlighted the need for new strategies to improve the reproducibility and comparability of Brillouin data: the EOM-BM could thus represent a significant innovation in this field.

Here, we assessed the temporal stability of the EOM-BM by continuously acquiring Brillouin data on water for over 50 hours, demonstrating superior spectral precision than a standard BM and enabling accurate, automated reconstruction of the temperature dependence of water sound velocity and attenuation.

We also performed long-term measurements in live cells expressing different SGs to study their Brillouin shifts. The stability of the EOM-BM was crucial for consistent mapping of mechanical properties across extended experimental sessions, overcoming common issues found in standard BMs such as temporal drifts and day-to-day variability, hindering the stability and repeatability of Brillouin measurements. This allowed us to generate accurate mechanical maps of condensates under physiological conditions and to discriminate between normal and pathogenic assemblies. Notably, we observed abnormal Brillouin shifts in disease-linked protein fragments, aligning with extensive literature showing their insolubility and cytotoxicity[53,61,63-68,71-77].

To validate our Brillouin shifts data, we performed parallel FRAP measurements. Interestingly, we observed a power-law dependence between the Brillouin shifts and the FRAP immobile fractions, that suggests an underlying percolation-like behavior[4,13] and supports the presence of fractal architectures within protein condensates, consistent with gel-like network transitions[4] observed in soft matter systems. These results represent an important experimental evidence of the fractal nature of SGs in vivo through a fully automated workflow. In this study, FRAP data were obtained by photobleaching the entire condensate and the resulting recovery curves were analyzed using single or double-exponential models, in line with the current standard in the field of biomolecular condensates[82,97]. A future extension of this work will include more advanced experimental[98] and analytical[78] FRAP methodologies to provide deeper insight into the diverse biophysical mechanisms of SGs organization.

With the adequate spectral resolution and temporal stability, our data demonstrate that Brillouin Microscopy can uncover intrinsic properties of SGs inaccessible to standard methods as FRAP. First, it can be applied to the characterization of biomolecular condensates in fixed samples: this capability would be of particular relevance for medical applications regarding the mechanical characterizations of patient-derived post-mortem tissues. Second, its unique label-free property allows to successfully assess the mechanical properties of the entire condensate compartment containing multiple protein species and thus distinguish between physiological and aberrant condensates, including those formed by pathogenic $FUS^{P525L}$, a known driver of aggressive ALS[16]. Beyond the ALS and FTD cases examined here, the label-free and noninvasive nature of Brillouin Microscopy might also be applied to other membrane-less organelles and protein condensates implicated in diverse proteinopathies (e.g., Huntington's, Parkinson's, Alzheimer's Diseases), regardless of their molecular composition.

Understanding how alterations in material properties of specific SGs contribute to severe neurological diseases remains one of the most critical unanswered questions in mechanobiology[2,4,14]. In conclusion, our study paves the way for the investigation of mechanical properties of biomolecular condensates in living cells with an automated, stabilized, high-precision Brillouin Microscope.

## Methods
This study did not involve human participants or animal subjects. All experiments were performed using established cell lines and did not require ethical approval.

### 532 nm Brillouin Microscope equipped with EOM
Our custom-built 532 nm confocal Electro-Optic Modulator (EOM)-equipped Brillouin Microscope (described in Fig. 1C) consists of an inverted microscope coupled to a double VIPA-based spectrometer through single-mode optical fibers[23].

The laser source is a continuous wave, single-mode, tunable laser (20 GHz around nearly 532 nm wavelength, Oxxius), fiber-coupled and then re-collimated in free space (element $a$ of Fig. 1C). Most of the laser signal is sent to the sample, while a small part is picked up by a beam splitter (BS, element $c$ of Fig. 1C) and coupled through a collimator ($b$) into a fiber-integrated EOM ($e$). The EOM is a phase-modulator (Jenoptik PM532) modulated by an amplified microwave generator (element $f$, which is a custom compact RF synthesizer by Lytid with a frequency stability of ± 25 ppm), coupled to a Mini Circuits ZVE-3W-183+ amplifier optimized in the 6-18 GHz frequency range; in such a manner, efficient modulation can be achieved up to 15 GHz using around 20 dBm of input RF power and achieving around 75% of modulation depth.

The EOM output is re-collimated and injected by a polarizing beam splitter (PBS, $c$) into the collection fiber. When modulating the EOM at a single frequency $\nu_{EOM}$, sidebands at the modulation frequency and at its harmonics, whose amplitudes depend on the power of the microwave source and are inversely proportional to the harmonic order, are added to the input monochromatic light. The EOM frequency is tuned by a custom-made MATLAB (Version R2024b) program.

The sample is placed on an inverted microscope (Olympus IX-73, placed in $d$). The 3D mechanical properties of the sample are recovered in a confocal laser scanning mode thanks to a pair of galvanometric mirrors (Thorlabs) and a z-piezo stage (MadCity Labs), already described elsewhere[29]. After passing through the PBS, the galvo mirrors and a quarter-wave plate, the laser beam is focused on the sample via an objective. Depending on the use, the objective is a 60X, 1.4NA, when imaging cells, or 4X, 0.11NA, when acquiring data on water. The Brillouin signal, scattered from a specific point of the sample, is collected by the same objective in a backscattering geometry to ensure minimum spectral broadening[39], and it is then coupled to the collection single-mode fiber. This fiber (core diameter=3.2 micron) has the double role of cleaning the spatial mode sent to the spectrum analyzer and assuring the confocality of the Brillouin measurement.

The collection fiber sends both the Brillouin signal and the EOM-modulated one to the spectrometer.

The spectrometer is composed of two orthogonal VIPAs (Light Machinery), both of Free Spectral Range (FSR) equal to 30 GHz. At the end of the spectrometer, an EMCCD camera (Evolve 512 Delta, Teledyne Photometrics) acquires the Brillouin spectrum. We adjusted the VIPA angle to allow the transmission of a couple of adjacent dispersion orders (sketched in Fig. 1A): when the spectrometer is correctly aligned, the Brillouin spectra show 2 Brillouin peaks (a Stokes and Anti-Stokes signal per order) of equal intensity.

While performing automatic pixel-to-GHz calibration via the EOM arm of the system, an automatic shutter (element $g$ of Fig. 1C), controlled by an external trigger, prevents the laser from reaching the

sample. This avoids its over-exposure due to the laser and interference to the calibration procedure from its Brillouin signal.

Aside from the Brillouin laser-scanning module, a standard brightfield and spinning disk fluorescence unit (XLIGHT V1, Crest Optics) is aligned at the microscope side port to retrieve morphological and fluorescence information from the sample, ensuring the match between the fluorescence and the Brillouin imaging planes.

The setup is controlled in MATLAB via a custom-built graphical user interface that, together with data acquisition and image reconstruction, also governs the EOM control loop, performs automatic pixel-GHz calibrations and executes the feedback control in a closed loop on the laser frequency.

The spectral precision of the microscope was measured as the standard deviation of the distribution of water Brillouin shifts, measured with a 4x objective (0.11NA), 100 ms exposure and 10 mW power: the obtained precision value was always between 8 and 10 MHz.

### 532 nm Brillouin acquisition of water in time

We acquired time lapses of a water sample over a long period (a total of 60' in Supplementary Fig. 1A, 100' in Fig. 3A, 52 h in Fig. 3B, 17 h in Supplementary Fig. 3) with a 4x objective (NA = 0.11, Olympus) to ensure minimal broadening and shift of the Brillouin peaks[39] (whose dependence from the objective NA is shown in Supplementary Fig. 2). The water sample was always maintained at a constant temperature of 27 °C by using a top-stage incubator (Okolab); while acquiring Brillouin data, two PT100 were used as temperature sensors in the sample and in the air.

All the data acquisitions followed the same protocol apart from the optional EOM control loop (performed in Fig. 3B and Supplementary Fig. 4) and calibration (performed in Fig. 3B). Briefly, at time = 0 we manually aligned the spectrometer on the water signal, thus ensuring that its Stokes and Anti-Stokes Brillouin peaks had the same intensity. Automatic pixel-to-GHz calibration with different EOM frequencies (centered at $\nu_{EOM} = 7$ GHz, 8.7 GHz, 10.9 GHz and 11.5 GHz, as described in Fig. 2) was then performed and 500 spectra of water were acquired with an exposure time of 100 ms per point and an optical power on the sample plane of 10 mW, for a total of ~1' acquisition. The acquisition was then paused for T = 2' (Fig. 3A), 10' (Supplementary Figs. 1A and 4) or 15' (Fig. 3B); while waiting, the shutter of Fig. 1C was automatically closed with an external trigger, so that the laser did not heat the sample. At time = T, with the shutter closed, in data shown in Fig. 3B the EOM control loop was performed on the Anti-Stokes EOM peak (centered at $\nu_{EOM} = 13$ GHz): if it moved more than 50 MHz from the initial one, the laser frequency was tuned accordingly (until their difference was lower than 50 MHz); if not, the laser frequency was not moved. This procedure ensured that the EOM was always centered at $\nu_{EOM} = 13.00 \pm 0.05$ GHz (as can be seen in temporal graphs of Figs. 3B, C and Supplementary Fig. 4). After this, automatic calibration with known EOM frequencies was performed. Then, the shutter was opened and a new acquisition of 500 water spectra began.

This scheme was automatically repeated in a time-lapse mode for N iterations (N = 6 for data in Supplementary Fig. 1A, N = 50 for Fig. 3A, and N = 208 for Fig. 3B).

### 780 nm Brillouin Microscope water data acquisition

The custom-built 780 nm confocal Brillouin microscope consists of an inverted Nikon Ti2 microscope that uses a continuous-wave 780 nm laser source (Toptica), locked on a Doppler-free absorption line of Rubidium-85 isotope. A custom-made optical filter[99] consisting of a Fabry-Perot (FP) crystal cavity and a Bragg grating is used to spectrally clean the incident light from amplified spontaneous emission (ASE) noise and spurious modes of the laser cavity. An optical sampler picks up a small portion of the light after the filter and sends it into a photodetector, and the optical power is monitored as feedback for stabilizing the temperature of the filter. The beam is then cleaned with a

spatial filter (10-micron pinhole) which is placed within a beam expander, used to ensure the filling of the objective pupil.

The sample is placed on a xy piezo stage (Prior), and the Brillouin signal is collected in a backscattering geometry through the same objective and coupled into a single-mode fiber, allowing for confocal sectioning. Unlike the green setup, here the Brillouin spectrometer is based on a single VIPA setup (FSR = 15 GHz, Light Machinery), where the elastic scattering light is suppressed by a Rubidium gas cell of 7.5 cm. The setup is controlled in MATLAB via a custom-built graphical user interface.

We acquired water data over a long period (a total of 270' in Supplementary Fig. 1B) with a 10x objective (0.11NA) to ensure minimal broadening and shift of the Brillouin peaks. As in the 532 nm experiment, water was maintained at a constant temperature of 27 °C within a stage-top incubator (Okolab); while acquiring Brillouin data, two PT100 were used as temperature sensors in the sample and in the air.

At time = 0 we manually aligned the spectrometer on the water signal, ensuring Stokes and Anti-Stokes Brillouin peaks had the same intensity. We then performed pixel-to-GHz calibration by locking the laser at different Rubidium absorption frequencies (thus ensuring a reference-free calibration as in the 532 nm Brillouin Microscope) and we acquired 500 spectra of water with an optical power on sample plane of 65 mW and an exposure time of 100 ms per point, for a total of ~1' acquisition. We then paused the acquisition and waited for T = 10' to cyclically perform another one, for a total of 27 acquisitions.

### Acquisition of water Brillouin data at different temperatures

We implemented a custom-built MATLAB code in our EOM-BM data acquisition routine that governed the temperature of the top-stage incubator (Okolab) and reached the setpoint with a tolerance of ±0.2 °C. Typically, ~40'−50' were needed for reaching every setpoint. After the setpoint has been reached, the routine automatically performed the EOM-control loop, pixel-to-GHz calibrations and acquired 500 Brillouin spectra; then, the acquisition moved to the next temperature setpoint. The acquisition lasted ~12 h.

We used a 4x, NA = 0.11, objective to ensure minimal spectral broadening[39].

While acquiring Brillouin data, two PT100 were used as temperature sensors in the sample and in the air.

We then transformed the Brillouin shift ($\nu_B$) in sound velocity ($V_S$) and the Brillouin width ($\Gamma_B$) in longitudinal kinematic viscosity ($v_L$) by using the following equations[20]:

$$V_S = \frac{\nu_B * \lambda}{2 * n(T)}$$

$$v_L = \frac{\Gamma_B}{8\pi} * \left(\frac{\lambda}{n(T)}\right)^2$$

Where n(T) is the index of refraction, whose dependency from temperature T is known[45]. The sound velocity values of water at different temperatures are also known with high precision[38] and this curve has been used as a reference.

### Brillouin spectra fitting

We fitted Brillouin signals by using the convolution between two functions: (i) the sum of two Lorentzian functions (for the Brillouin peaks) and two Gaussian functions (for the EOM peaks), and (ii) the experimental spectrometer line shape, extracted during the EOM calibration step. In this way, retrieved Brillouin FWHMs were not affected by VIPAs broadening.

### Cell culture preparation and maintenance

SK-N-BE cells (purchased from ATCC, catalog number CRL-2271) were cultured in RPMI (Sigma-Aldrich, Saint Louis, MO, USA) supplemented

with 10% FBS (Sigma-Aldrich, #F2442), GlutaMAX supplement 1X (Thermo Fisher Scientific, # 35050061), sodium-pyruvate 1 mM (Thermo Fisher Scientific, #11360070) and Pen/Strep 1X (Sigma-Aldrich, #P4458). For the inducible expression of SOD1^WT or SOD1^A4V, SK-N-BE cells were exposed to 10 ng/mL Doxycycline (Sigma-Aldrich, #D9891) for 48 h. For the inducible expression of TDP-43 C-ter fragments, SK-N-BE cells were exposed to 200 ng/mL Doxycycline for 72 h before the sodium arsenite treatment. For the inducible expression of FUS^P525L, SK-N-BE cells were exposed to 50 ng/mL Doxycycline for 24 h before the sodium arsenite treatment. For the inducible expression of EGFP alone, SK-N-BE cells were exposed to 200 ng/mL Doxycycline for 24 h. For the oxidative stress induction, cells were treated with 0.5 mM sodium arsenite (Sigma-Aldrich, #106277) for 1 h.

Cells were passed when reaching ~70–80% confluency. For Brillouin Microscopy and FRAP, cells were seeded into 8-well ibiTreat mslides (ibidi) at low density; the next day cells were live imaged.

## Plasmids construction and transfection
The epB-Puro-TT-TDP-43-CTF-209-EGFP and epB-Puro-TT-TDP-43-CTF-220-EGFP were generated by inserting the transgene sequences into the enhanced piggyBac transposable vector[100]. The TDP-43 C-Terminal Fragments (CTFs) were cloned between the BamHI and NotI sites of the epB-Puro-TT vector[100]. The EGFP sequence was obtained from pEGFP-N1 (Clontech) by cutting the plasmid with HindIII and NotI enzymes. The resulting constructs contain the enhanced piggyBac terminal repeats flanking a constitutive cassette driving the expression of the puromycin resistance genes fused to the *rtTA* gene and, in the opposite direction, a tetracycline-responsive promoter element (TRE) driving the conditional expression of the transgenes.

The FUS^P525L expressing construct is described elsewhere[101].

The plasmid containing a C-terminal GFP-tagged form of human SOD1^WT or SOD1^A4V was a gentle gift from Prof. Mariangela Morlando's Lab (Sapienza University of Rome). The either WT or A4V sequence of SOD1-GFP was subcloned in the epB-Puro-TT vector[100]; briefly, SOD1-GFP was PCR-amplified with CloneAmp HiFi PCR Premix (TakaraBio) and cloned in the acceptor plasmid linearized by PCR using the InFusion Cloning kit following manufacturer protocol (TakaraBio); the oligonucleotides employed are listed below. The EGFP alone expressing construct is described elsewhere[101]. Stable cell lines were generated taking advantage of the transposon-based piggyBac technology. SK-N-BE cells were co-transfected with a 1:10 ratio of the piggyBac transposase and the transposable vector using Lipofectamine 2000 (Life Technologies) following the manufacturer's instructions. Selection with puromycin resulted in stable and inducible cell lines, which were then tested for consistent expression of the fluorescent protein. SK-N-BE cell lines expressing the 3xFlag-tagged-FUS^P525L under a doxycycline-inducible promoter were already present in the lab[87].

The list of oligonucleotides employed, together with their name and 5′-3′ sequence, is: ePB INV FW: TGCGGCCGCGACTCTAGATC; ePB INV Rev: TACCGAGCTCGAATTCTCCAGG; SOD1 WT Inf FW: AATTCGAGCTCGGTAATGGCGACGAAGGCCGTGTG; SOD1 A4T Inf FW copy: AATTCGAGCTCGGTAATGGCGACGAAGGCCGTGTG; SOD1 A4T Inf FW copy: AATTCGAGCTCGGTAATGGCGACGAAGGTCGTGTG; EGFP InF Rev: AGAGTCGCGGCCGCATTACTTGTACAGCTCGTCCATG.

## Immunofluorescence
Cells growing on coverslips were fixed for 20 min at RT with cold 4% paraformaldehyde (Electron Microscopy Sciences, #15710) diluted to in complete PBS (Sigma-Aldrich, #D1283), rinsed 3 times with complete PBS and stored in PBS at 4 °C. Cells were permeabilized with 0.3% Triton X-100 (Sigma-Aldrich, #648466) diluted in complete PBS for 10 min and blocked for 30 min at RT with 5% goat serum. Samples were then incubated overnight at 4 °C with the primary antibodies rabbit anti-G3BP1 (1:300, ab181150 Abcam) and mouse anti-FLAG (1:400, #F1804 Sigma-Aldrich) diluted in blocking solution (5% goat serum in

PBS, #G9023 Sigma-Aldrich). Cells were washed with complete PBS three times for 5 min at RT and then incubated with the secondary antibodies goat anti-rabbit Alexa Fluor 488 (1:300, #A11008 Thermo Fisher Scientific) and donkey anti-mouse Alexa Fluor Plus 647 (1:300, #A32787 Thermo Fisher Scientific) for 1 h at RT diluted in blocking solution and were incubated for 1 h at RT. Nuclei were stained with 1 μg/ml DAPI (Sigma-Aldrich, #D9542) diluted in complete PBS for 5 min and coverslips were mounted applying ProLong™ Glass Antifade Mountant (Thermo Fischer Scientific, #P36980) leaving the slides overnight on the bench, covered from light. Confocal images were acquired with an inverted Olympus iX73 equipped with an X-Light V3 spinning disc head (Crest Optics), a Prime BSI Scientific CMOS (sCMOS) camera (Photometrics), a LDI laser illuminator (89North) and MetaMorph software (Molecular Devices), as Z-stacks (0.3-um step size) with a 60×, NA 1.42, oil-immersion objective (Olympus).

## Brillouin acquisitions of living and fixed cells
24 hours prior to Brillouin live imaging, cells were seeded into 8-wells chambers (ibidi) and left in their culture media. We stained cells for 20′ with 2 drops/ml of Hoechst 33342 (Thermo Fisher Scientific, #R37605) and imaged them under transmission and fluorescence mode.

We took advantage of our top-stage incubator (Okolab) to have stable temperature, humidity and $CO_2$ for live cells imaging. For Brillouin data acquisitions we used a 60x, 1.40 NA (Olympus), to ensure high resolution mechanical mapping of the cells. The longitudinal step size on the sample was 300-400 nm, the acquisition time was 60-70 ms per point, and the optical power of the Brillouin laser delivered to the specimen was 6-8 mW.

Fixed cells were treated with 4% PFA for 15′ and left at 4 °C. We then imaged them using the same parameters, but with a laser power of 20 mW on the sample plane and a reduced exposure time of 30-40 ms per point.

No other manipulations have been applied for visualization of Brillouin maps. All data analysis has been performed using custom-made MATLAB codes.

## FRAP data acquisitions and analysis
FRAP data were obtained by photobleaching a specific region of the cells within the GFP-labeled condensates and monitoring its intensity recovery over time. These data were always acquired using the same batch of cells and in the same experimental day as the Brillouin imaging, on another microscope equipped with a FRAP module.

Briefly, we used an inverted iX83 FV1200 Olympus laser scanning microscope equipped with a 60x, NA 1.35, oil objective (Olympus) and a cellVivo incubation system (PeCon) for controlling temperature, humidity and $CO_2$ concentration. We adjusted image size and pixel (zoom 3x, 500×256 px) and scanning at maximum speed to allow 560 ms/frame scanning time and run a time lapse streaming acquisition of 300 frames. For the FRAP protocol we acquired first 3 images as pre-bleaching reference, then on the 4th scan we coordinated the main scanner with the SIM scanner in the Tornado mode to bleach for 50 ms a round region of 30 pixels in diameter (1 px = 0.174 μm, thus obtaining a ROI of 5.22 microns in diameter) using the 473 nm laser at 0.92 mW power, and finally following the recovery for the remaining frames. For each sample we performed 3 replicates with at least 10 FRAP each.

All FRAP data were analyzed with custom-made MATLAB codes that normalized the intensity of every region and monitored its behavior in the different frames. These curves were fitted with custom-made MATLAB scripts by using non-linear least squares algorithms. Depending on the recovery, we applied either single-exponential (for GFP, pure diffusion case, and TDP-43^220-GFP due of its limited fluorescence recovery) or double-exponential (for all the other proteins, reflecting both diffusion and binding dynamics) models[21,78,79].

Information about microscopy images obtained with FRAP are reported in the Light Microscopy reporting table (Supplementary Data 1).

## Statistical analysis and Reproducibility

For all experiments on living cells, we always had at least 3 independent biological replicates. All statistical analysis has been performed using custom-made MATLAB codes. We always checked the normality of data with a Kolmogorov–Smirnov test and then performed either two-sided t-tests (if data were normally distributed) or Wilcoxon rank-sum tests (if not). In case of multiple comparisons, we performed two-sided one-way ANOVA (if data were normally distributed) or Kruskall-Wallis tests, followed by multiple comparison post-hoc tests. A p-value of $p < 0.05$ was chosen as statistically significant.

All the results shown in the graphs are given as mean ± standard deviation (SD) or as mean ± 2*standard error of the mean (S.E.M.); confidence intervals for the fits are 95% unless otherwise specified. The confidence intervals of the power-law fit[102–106] shown in Fig. 6 have been obtained with OriginLabPro 2025b, while all the others with MATLAB.

All representative spectra or datasets derive from independent experiments, all of which were repeated several (i.e., at least 10) times and produced highly consistent and robust results.

## Reporting summary

Further information on research design is available in the Nature Portfolio Reporting Summary linked to this article.

## Data availability

All the data generated in this study are provided in the Source Data file and are available in the Figshare database[107] under accession code [https://doi.org/10.6084/m9.figshare.30316738]. Source data are provided with this paper.

## Code availability

Custom MATLAB codes used for this study are publicly available in the ZENODO database[108] at: (https://doi.org/10.5281/zenodo.18187110).

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

## Acknowledgements

This research was funded by grants from ERC-2019-Synergy Grant (ASTRA, n. 855923); EIC-2022-PathfinderOpen (ivBM-4PAP, n. 101098989); the National Center for Gene Therapy and Drugs Based on RNA Technology, National Recovery and Resilience Plan (NRRP), Mission 4 "Education and Research", Component 2 "From Research to Business", Investment 1.4 "Strengthening research structures for supporting the creation of National Centers, national R&D leaders on some Key Enabling Technologies", funded by the European Union - Next Generation EU, Project CN00000041, CUP B93D21010860004 and CUP J33C22001130001, to I.B, A.R. and G.R.; POR FESR Lazio 2014-2020 Azione 1.2.1 (SIMBA, n. A0375-2020-36389) to A.R. and C.T.; Sapienza departmental projects 2023 (RD12318A998C70B8) to G.D.T.; Marie Skłodowska-Curie Action (MSCA) under the Horizon Europe Program for doctoral education and postdoctoral training of researchers (FaBriCA-Tion, n. 101103038) to L.Z. The authors wish to thank the staff of the technical office and the microscopy facility of CLN²S for technical support.

## Author contributions

C.T. wrote the manuscript with valuable revision from all the authors. C.T., E.P., A.R. and G.R. conceived the study. E.P. developed the EOM method and implemented it in the optical setup, with the help of G.Z.; F.G., G.Z. and C.T. developed the EOM control loop software. C.T. acquired the water data on the 532 nm Brillouin Microscope, performed the formal analysis, interpreted the results and produced all the figures. G.R. interpreted the theoretical model and fitted the percolation data. C.B. and C.M. performed all the measurements on cells at the Brillouin Microscope (with supervision of C.T.). L.Z. and N.A. performed the acquisitions of water data at 780 nm Brillouin Microscope. A.R. and

G.d.T. conceived the biological application on biomolecular condensates. M.G.G., A.G., and G.d.T. developed the cellular model and interpreted the biological results, with supervision of I.B. and A.R. C.M. maintained the cells in culture. A.G. performed immunofluorescence data. V.d.T. performed FRAP measurements and interpreted the results. C.B. and F.G. developed the custom MATLAB code to fit FRAP data. G.R. supervised the project. All authors reviewed the paper.

## Competing interests

E.P., F.G., and G.Z. are employed by CREST Optics S.p.A. The other authors have no conflicts of interest to disclose.
