## [Transparent Peer Review file · Nature Communications]

Stabilized real-time Brillouin microscopy reveals fractal organization of protein condensates in living cells

Corresponding Author: Dr Claudia Testi

Version 0:

Reviewer comments:

Reviewer #1

(Remarks to the Author)

The paper introduces a novel method to calibrate a Brillouin microscope, based on the use of a pilot tone injected into the spectrometer through an external electro-optic modulator. The same method is used to compensate for temperature drifts of the spectrometer and frequency drifts of the laser source. The method is convenient as it avoids the need for frequent manual realignments of the spectrometer and repeated calibrations over time. Furthermore, the authors demonstrate that, using the Brillouin shift of known substances like water or methanol as calibration references is ineffective, as it varies with temperature and microscope NA. The effectiveness of the optimized microscope has been demonstrated by measuring the mechanical properties of various protein condensates in living cells, and validating their findings with FRAP measurements. The paper is well written and deserves publication. We have only a few minor concerns:

- 1) The proposed method relies on the spectral line generated by the EOM to compensate for any drift due to laser or environment. A brief discussion about the advantage of using that peak instead of the Rayleigh spectral line would be helpful. We realize that the Rayleigh scattering could be too strong for this purpose. Nonetheless, one may consider using a filter to attenuate the Rayleigh backscatter, as demonstrated in various works (e.g. P. Shao et al. "Etalon filters for Brillouin microscopy of highly scattering tissues," *Opt. Express* 24, 22232-22238 (2016)).
- 2) Lines 80-83: The authors state that: "This spectrum (...) arises from the scattering of light upon interaction with the sample's thermally activated sound waves: it consists of a central Rayleigh peak, due to light elastically scattered and two symmetric side peaks (the Brillouin peaks)". That statement may be slightly misleading, as the Brillouin spectrum only includes the two side peaks. The authors may revise this statement by referring to the spectrum in Fig. 1A as the "backscattered spectrum", which includes the Rayleigh (elastic) component and the Brillouin scattering spectrum.
- 3) Line 594. Please provide the model and manufacturer of the EMCCD camera.
- 4) Lytid seems to provide only subTHz sources (starting from 60 GHz), Please check the manufacturer or provide the model number.
- 5) Line 570. Please provide the RF power used to drive the modulator, along with the corresponding modulation depth.
- 6) Please provide more information about the accuracy and stability of the RF generator driving the EOM modulator.
- 7) It is stated that a threshold of 50 MHz was used to trigger the tuning of the laser frequency. Please clarify the criteria used to select this value.

Reviewer #2

(Remarks to the Author)

Reviewer #3

(Remarks to the Author)

The manuscript is very interesting. It presents the development of the Electro-Optic Modulator Brillouin Microscope and

demonstrates its application to investigate the internal molecular organization of biomolecular condensates. Condensates undergoing liquid to solid transitions exhibit structural changes, and consequently different mechanical properties. EOM-BM advantages are clear and well introduced: its development is impressive, and fits well within the general need for microscopes capable of providing structural and potentially mechanical characterization of biological structures in vivo. The manuscript is clearly written, and well organized. My concerns are as follows.

1) FRAP is the standard method to assess molecular mobility within condensates in cellulo. Indeed, other techniques, such as passive microrheology, coalescence observations, Atomic Force Microscopy (AFM), and optical tweezers, are used only in vitro. However, these latter ones directly measure mechanical responses while FRAP does not directly probe condensates mechanics, rather their molecular mobility, and has several well-recognized limitations. For instance, the authors treat FRAP recovery curves with single or double-exponential models, the latter suggesting the presence of both fast and slow diffusion components. However, these models do not differentiate between various underlying mechanisms, such as internal diffusion within the condensate, importantly the exchange with the surrounding dilute phase, or the formation of internal clusters. The use of half-FRAP (doi: 10.1038/s41467-022-35430-y) to assess condensates internal dynamics would have been more appropriate for comparison with EOM-BM.

In contrast, EOM-BM has one clear and important advantage over FRAP, as reported in Figures 7 and S8. Indeed, it "... probes the mechanical properties of the entire local compartment rather than merely the tagged protein..." This is a major strength compared with all fluorescent-based techniques and I would have expected more emphasis on that, such as its inclusion in the abstract.

2) The authors do not report laser power (in Watt units) and exposure time used in FRAP acquisitions. Some condensates may gelify upon intense and long FRAP exposure, for example if proteins are thermosensitive. Did the authors compare fluorescence recovery under different laser powers or exposure times to exclude FRAP-induced structural variations, potentially affecting data in Figures 5 and 7? Moreover, FRAP ROIs were chosen to be 3–4 μm in size, occasionally resulting in ROIs larger or smaller than the exposed condensates. This size mismatch can affect the observed diffusion dynamics and the balance between internal diffusion and exchange with the surrounding dilute phase. How did the authors account for this effect in their analysis?

3) As for the FRAP case, the EOM-BM characterization of biomolecular condensates presented in the manuscript is mainly structural. The study does not provide any quantified mechanical property for condensates. I believe that the terms "mechanical characterization" and "quantified mechanical properties" are overused throughout the manuscript and in the abstract. In this regard, why the authors did not extrapolate any viscosity values from the condensates dataset?

MINOR:

1) Several figures have heterogeneous font size within the same panels. In some cases the text is small and it is hardly readable. Graphs within the same figures and panels can differ in size and could be better aligned/organized. FRAP ROIs highlighted by the red circles in Figure S6 are barely visible.

2) Figure 2B: why 14 points in the upper graph (calibration)? Shouldn't we expect 16 points?

3) Literature 1 (intro): I share with the authors the emergency of developing instruments capable of investigating "in vivo 3D biological environments" with high spatial and temporal resolution. Their development fits well within this context, but the authors did not discuss efficient and faster techniques: in vivo imaging with fluorescence lifetime probes can quantify membrane tension (e.g. doi: 10.1021/jacs.8b13189), or local viscosity (e.g. Bodipy probes).

4) Literature 2 (intro): the authors did not discuss AFM, which remains the principal technique for characterizing the mechanics of liquid–solid phase transitions in biomolecular condensates, yet in vitro. AFM, like Brillouin microscopy, is label-free and complementary, as it characterizes mechanical properties at high temporal resolution, as in microrheology, quantifying Young's moduli, surface tensions, and viscosities (doi: 10.1016/j.xcrp.2025.102430, and 10.1073/pnas.2304714120).

Reviewer #4

(Remarks to the Author)

This manuscript presents the results of a study conducted on stress granules (SGs) in living cells using a Brillouin microscope stabilized using an electro-optic modulator (EOM). The use of EOMs in Brillouin spectrometers has previously been demonstrated (ref. 36), however not in VIPA-Brillouin systems to the best of my knowledge. So there is some novelty in this application.

The results are notable, especially in the presence of large temperature fluctuations ($\sim 2.5^\circ\text{C}$; Supplementary Information).

Comments/questions:

- it is not clear how the deconvolution of the Brillouin spectra is performed (Brillouin Spectral Fitting) between having a Lorentzian-shape Brillouin signal and a Gaussian EOM signal.
- it is not clear what is the composition of the SGs - what if those are lipid droplets or else? How specific is the staining for validation?
- how generalisable are these results to other proteinopathies?
- how translatable are these findings to medical applications of Brillouin microscopy?

Version 1:

Reviewer comments:

Reviewer #1

(Remarks to the Author)

All my comments have been properly addressed. I am satisfied with the authors's replies. I recommend publication of the manuscript in its present form.

Reviewer #2

(Remarks to the Author)

Reviewer #3

(Remarks to the Author)

The revised manuscript is now improved and is essentially ready for publication.

I am still convinced that in some parts of this manuscript the mechanical properties are described in terms that lies between qualitative and quantitative description. However, after the authors' corrections, this is now only a semantic issue, beyond the scope of the manuscript.

Last: the statement "AFM spatial resolution is also limited: despite the use of a very small tip, its resolution on biological materials generally exceeds several times the tip size" is fundamentally incorrect.

Hundreds of publications demonstrate that AFM resolution on biological materials can be comparable to or even better than the probe size, ranging from biological membranes (doi: 10.1073/pnas.0437992100) to biomolecular condensates in vitro (doi: 10.1038/s41586-020-1977-6). I am providing these dois as examples, not as suggestions for inclusion, obviously. The authors' claim applies only to images acquired at the apical membrane of some eukaryotic cells, where, in any case, AFM resolution does not exceed several times the tip size.

That said, I confirm that the manuscript is very interesting, clearly written and organized.

Response to Reviewers

We thank the reviewers for their careful evaluation of our manuscript and their constructive feedback. We also thank them for recognizing the significance and thoughtful design of our study. Their comments have been important in enhancing the quality and impact of our work: in response, we have made a concerted effort to address each of their questions and to provide comprehensive clarifications where needed.

*In the revised manuscript all changes have been highlighted in yellow. We have incorporated all the reviewers' comments and suggestions. Furthermore, we now provide full access to the underlying numerical data for all figures and charts through a newly added **Supplementary Data 1**, which contains the complete raw dataset in Excel format.*

*Please find attached a detailed, point-by-point response to all reviewer comments. Reviewer remarks are presented in **bold**, with our responses in italics. All line references correspond to the revised version of the manuscript.*

Reviewer #1:

The paper introduces a novel method to calibrate a Brillouin microscope, based on the use of a pilot tone injected into the spectrometer through an external electro-optic modulator. The same method is used to compensate for temperature drifts of the spectrometer and frequency drifts of the laser source. The method is convenient as it avoids the need for frequent manual realignments of the spectrometer and repeated calibrations over time. Furthermore, the authors demonstrate that, using the Brillouin shift of known substances like water or methanol as calibration references is ineffective, as it varies with temperature and microscope NA. The effectiveness of the optimized microscope has been demonstrated by measuring the mechanical properties of various protein condensates in living cells, and validating their findings with FRAP measurements.

The paper is well written and deserves publication.

We are grateful to the reviewer for acknowledging the significance and careful design of our study. We hope that our responses to the minor points raised will help improve the clarity and quality of the manuscript.

We have only a few minor concerns:

1) The proposed method relies on the spectral line generated by the EOM to compensate for any drift due to laser or environment. A brief discussion about the advantage of using that peak instead of the Rayleigh spectral line would be helpful. We realize that the Rayleigh scattering could be too strong for this purpose. Nonetheless, one may consider using a filter to attenuate the Rayleigh backscatter, as demonstrated in various works (e.g. P. Shao et al. "Etalon filters for Brillouin microscopy of highly scattering tissues," *Opt. Express* 24, 22232-22238 (2016)).

We thank the reviewer for raising this point. As correctly noted, our approach is using the EOM-generated peaks as reproducible, stable, programmable references, replacing the Rayleigh signal.

*While in principle a filtered Rayleigh peak could also serve as a reference, implementing this solution in an optical setup poses several technical challenges. First, efficient suppression requires filters with very high extinction (>70 dB), which is technically demanding and comes at the cost of significant optical losses and delicate beam alignment. Free-space interferometric filters (e.g., Michelson interferometers as in Antonacci et al., *Appl. Phys Lett.* 2015, doi: 10.1063/1.4927400) could be used for this purpose and, indeed, our lab used such*

a filter in the past to partially suppress Rayleigh signals (see Martinez-Vidal, Testi et al., Scientific Reports 2024, doi: 10.1038/s41598-023-51006-2). However, the performance of these free-space filters is highly sensitive to mechanical vibrations and temperature: small temperature drifts, as the 1–2 °C variations we show in our work, can randomly occur over the course of a day, even in a controlled temperature laboratory. Such instability results in frequent (i.e. every 5-10 minutes) manual adjustments of the filter to suppress Rayleigh signal in turbid media such as cells, thus limiting its suitability for long-term, user-independent measurements, which was the main point of our manuscript. If the filter is not very finely tuned during time, indeed, Rayleigh signal saturates and overcomes Brillouin peaks.

Moreover, even if the filter stabilization is successful, the residual Rayleigh signal often exhibits strong intensity fluctuations over time, ranging from almost saturation to complete disappearance depending on the location of the beam on the sample. This variability complicates the spectral fitting procedure and introduces additional sources of error. The work cited by the reviewer indeed highlights the potential of etalon filters, but even there the temporal stability was maintained only for ~1 hour within ± 1 dB, which is not sufficient for the multi-day temporal stability targeted in our study.

In contrast, in a double-VIPA spectrometer design, which is the state-of-the-art (Zhang & Scarcelli, Nat Protocols 2021, doi: 10.1038/s41596-020-00457-2) for precise biomechanical Brillouin characterization, the Rayleigh peak is physically blocked; this avoids signal saturation and enhances Brillouin peaks contrast, but at the cost of the absence of a fixed internal reference. This is precisely the gap that the EOM signals fill.

The EOM-generated peaks offer several decisive advantages: they require no filtering, remain stable once aligned at the optical input, do not suffer from temporal fluctuations, and their position can be tuned to avoid spectral overlap with Brillouin peaks (something the Rayleigh line cannot provide, even after filtering, due to its broad tails). This results in a simpler optical setup, more robust data fitting, and most importantly, reliable long-term spectral stability. We therefore believe that the EOM-based approach represents the most practical and effective solution for achieving automated, stable, and reproducible Brillouin measurements over extended periods of time.

Following the reviewer's suggestion, we decided to expand this part in the Results section, remarking the instability in time of such filters for the target of our study (lines 552-558 of the revised version): "Although a filtered Rayleigh signal might, in principle, provide a reliable frequency reference, implementations based on free-space interferometric filters (such as Michelson interferometers^{91,92}) are highly sensitive to mechanical vibrations and temperature fluctuations: thus, these systems require frequent (i.e. every 5-10 minutes) manual adjustments to maintain effective suppression in turbid media such as living cells.

In contrast, EOM-generated peaks remain stable once aligned at the optical input and exhibit no temporal intensity fluctuations”.

2) Lines 80-83: The authors state that: “This spectrum (...) arises from the scattering of light upon interaction with the sample’s thermally activated sound waves: it consists of a central Rayleigh peak, due to light elastically scattered and two symmetric side peaks (the Brillouin peaks)”. That statement may be slightly misleading, as the Brillouin spectrum only includes the two side peaks. The authors may revise this statement by referring to the spectrum in Fig. 1A as the “backscattered spectrum”, which includes the Rayleigh (elastic) component and the Brillouin scattering spectrum. *We thank the reviewer for this comment. We rephrased the paragraph in the revised manuscript (lines 101-104): “This spectrum (Figure 1A, left panel) consists of a main central elastic peak (Rayleigh) at the incident light frequency, and two symmetric side peaks (Brillouin peaks), which arise from the inelastic scattering of light at lower (Stokes) and higher (anti-Stokes) frequencies upon interaction with the sample’s thermally activated sound waves.”*

3) Line 594. Please provide the model and manufacturer of the EMCCD camera. *We apologize for this missing information. The model of the EMCCD is Evolve 512 Delta, Teledyne Photometrics. In the reviewed version of the manuscript (line 657) we now provide this information: “At the end of the spectrometer, an EMCCD camera (Evolve 512 Delta, Teledyne Photometrics) acquires the Brillouin spectrum.”*

4) Lytid seems to provide only subThz sources (starting from 60 GHz) , Please check the manufacturer or provide the model number.

We apologize for this inaccuracy. Our Lytid RF synthesizer is a custom one, providing a frequency band between 10 MHz and 25 GHz and upgraded with a high-power amplifier (MiniCircuits ZVE-3W-183+) optimized to work in the 6-18 GHz frequency range. We now included this in the revised Methods of the manuscript (lines 629-633): “The EOM is a phase-modulator (Jenoptik PM532) modulated by an amplified radio-frequency driver (element f, which is a custom compact RF synthesizer by Lytid with a frequency stability of ± 25 ppm), coupled to a Mini Circuits ZVE-3W-183+ amplifier optimized in the 6-18 GHz frequency range; in such a manner, efficient modulation can be achieved up to 15 GHz using around 20 dBm of input RF power and achieving around 75% of modulation depth”.

5) Line 570. Please provide the RF power used to drive the modulator, along with the corresponding modulation depth.

The RF power of the custom synthesizer coupled with the amplifier can be varied between 13dBm and 33dBm, but we typically used around 20dBm, which corresponds to a modulation depth of about 75%. We now included this value in the revised version of the manuscript, in the Methods section (in the above-mentioned lines 629-633).

6) Please provide more information about the accuracy and stability of the RF generator driving the EOM modulator.

From the datasheet of the Lytid RF synthesizer, its frequency stability is ± 25 ppm. We now included this parameter in the revised text (lines 629-633). Unfortunately, the datasheet does not provide an accuracy value.

7) It is stated that a threshold of 50 MHz was used to trigger the tuning of the laser frequency. Please clarify the criteria used to select this value.

We selected this threshold as a compromise between two competing requirements during data acquisition: ensuring sufficiently frequent recalibrations to maintain spectral accuracy (thus avoiding large drifts that could distort the Brillouin spectra), while minimizing acquisition interruptions and measurement time loss. Through empirical testing, we found that setting the threshold at 50 MHz provided an optimal balance between these two necessities. In practice, this value resulted in automatic recalibrations approximately every 10–15 minutes, which was frequent enough to guarantee spectral stability without interrupting the acquisition workflow unnecessarily.

We inserted this clarification in the revised text (lines 251-255): “In our protocol, this threshold was set at 50 MHz as an empirical compromise between two competing requirements during data acquisition: ensuring sufficiently frequent recalibrations to maintain spectral accuracy and prevent Rayleigh line spillover, while minimizing acquisition interruptions and measurement time loss.”

Reviewer #2:

We thank the reviewer for this comment and hope they will appreciate the corrections we made to the text.

Reviewer #3:

The manuscript is very interesting. It presents the development of the Electro-Optic Modulator Brillouin Microscope and demonstrates its application to investigate the internal molecular organization of biomolecular condensates. Condensates undergoing liquid to solid transitions exhibit structural changes, and consequently different mechanical properties. EOM-BM advantages are clear and well introduced: its development is impressive, and fits well within the general need for microscopes capable of providing structural and potentially mechanical characterization of biological structures in vivo. The manuscript is clearly written, and well organized.

We thank very much the reviewer for all the positive comments and for recognizing the sound significance of our manuscript. We hope that the following responses address all the concerns raised and contribute to enhance the quality of the manuscript.

My concerns are as follows.

1) FRAP is the standard method to assess molecular mobility within condensates in cellulo. Indeed, other techniques, such as passive microrheology, coalescence observations, Atomic Force Microscopy (AFM), and optical tweezers, are used only in vitro. However, these latter ones directly measure mechanical responses while FRAP does not directly probe condensates mechanics, rather their molecular mobility, and has several well-recognized limitations. For instance, the authors treat FRAP recovery curves with single or double-exponential models, the latter suggesting the presence of both fast and slow diffusion components. However, these models do not differentiate between various underlying mechanisms, such as internal diffusion within the condensate, importantly the exchange with the surrounding dilute phase, or the formation of internal clusters. The use of half-FRAP (doi: 10.1038/s41467-022-35430-y) to assess condensates internal dynamics would have been more appropriate for comparison with EOM-BM.

We thank the reviewer for the opportunity to discuss this important point. We agree that FRAP does not directly probe condensate mechanics but rather their mobility. Nevertheless, mobility and mechanics are highly related: as reported extensively in the literature, higher molecular mobility has been consistently linked to liquid-like properties, while the opposite to solid/gel-like state. For this reason, FRAP has been used in several seminal studies as the standard method to indirectly infer biomolecular condensates mechanical properties, particularly to characterize the liquid-to-solid phase transitions (see, for example, Ray et al., Nature Chemistry 2020, doi: 10.1038/s41557-020-0465-9; Jain et al., Cell 2016, doi:

10.1016/j.cell.2015.12.038; Feric et al., *Cell* 2016, doi: 10.1016/j.cell.2016.04.047; Shin et al., *Science* 2017, doi: 10.1126/science.aaf4382; Lu et al., *Nature Cell Biology* 2022, doi: 10.1038/s41556-022-00988-8; and many others cited in our Reference section).

Thus, our intention was to benchmark our enhanced Brillouin Microscope against a well-established technique, whose use in literature has been proved to be solid in differentiating condensates mechanics. While we agree with the Reviewer that the Half-FRAP can offer additional mechanical insights, its use in literature remains less widespread than conventional full-FRAP. For example, even very recent important reviews on current practices for studying biomolecular condensates directly report FRAP as the standard method for condensate characterization, but do not mention half-FRAP (see for example Jeon et al., *Signal Transduction and Targeted Therapy* 2025, doi: 10.1038/s41392-024-02070-1). Indeed, half-FRAP is intended to be used for discriminating the differences in the internal dynamics between ICBS (low-valency interactions with spatially clustered binding sites) and LLPS phenomena, whereas in our study the goal was to assess phase state transitions, for which we believe that conventional FRAP is sufficient. That said, we agree that a natural extension of our work by using half-FRAP would be highly valuable for probing more detailed aspects of condensate organization, and we thank the reviewer for this insightful suggestion. Regarding data fitting, we employed mono-exponential or bi-exponential models, following established practice in the literature (Reits et al., *Nat Cell Biol* 2001, doi: doi.org/10.1038/35078615; Srikantha et al., *PLoS One* 2022, doi: 10.1371/journal.pone.0261925).

We would also like to remark that our positive control for LLPS was G3BP1, the core protein of stress granules that is well known to undergo LLPS (Yang et al., *Cell* 2020, doi: 10.1016/j.cell.2020.03.046). In our FRAP data (Figure 5C, second panel), indeed, the LLPS behavior of this protein was clearly distinguishable from controls, as the two distributions were statistically significant different. As a consequence, our internal control effectively validated the assay and there was no need for further distinctions in granules dynamics. In addition, data obtained on G3BP1-GFP (Figure 5C) and on G3BP1-GFP treated with DOXY (Figure 7A) were in agreement with independent data obtained with Fluorescence Correlation Spectroscopy (see ref #104: Mariani et al., *Nucleic Acid Research* 2024, doi: 10.1093/nar/gkae942, where the same cellular model has been used), a completely different methodology that does not rely on photobleaching, thus corroborating our FRAP results. Consequently, we believe that our FRAP analysis is appropriate and sufficiently robust for the purposes of this study.

In the Discussion of the revised manuscript, we included this statement as a future extension of our work (new lines 590-595): “In this study, FRAP data were obtained by photobleaching

the entire condensate and the resulting recovery curves were analyzed using single or double-exponential models, in line with the current standard in the field of biomolecular condensates^{85,93}. A future extension of this work will include more advanced experimental⁹⁴ and analytical⁸¹ FRAP methodologies to provide deeper insights into the diverse biophysical mechanisms of SGs organization”.

In contrast, EOM-BM has one clear and important advantage over FRAP, as reported in Figures 7 and S8. Indeed, it “...probes the mechanical properties of the entire local compartment rather than merely the tagged protein...” This is a major strength compared with all fluorescent-based techniques and I would have expected more emphasis on that, such as its inclusion in the abstract.

We thank the reviewer for highlighting this important aspect and we agree with this suggestion. We now include this phrase in the section Results (lines 523-526): “in particular, this technique enables to assess the properties of the entire condensate compartment containing multiple protein species, rather than being limited to the tagged protein of interest as in fluorescence-based approaches” and in the Discussion (lines 602-606): “Second, its unique label-free property allows to successfully assess the mechanical properties of the entire condensate compartment containing multiple protein species and thus distinguish between physiological and aberrant condensates, including those formed by pathogenic FUS^{P525L}, a known driver of aggressive ALS⁵⁴”. We also inserted it in the new abstract (new lines 28-32: “Furthermore, our data demonstrate that the label-free nature of Brillouin microscopy enables to probe the mechanical behavior of entire condensate compartments containing multiple protein species, overcoming the limitations of fluorescence-based methods that target only specific labeled proteins.”).

2) The authors do not report laser power (in Watt units) and exposure time used in FRAP acquisitions. Some condensates may gelify upon intense and long FRAP exposure, for example if proteins are thermosensitive. Did the authors compare fluorescence recovery under different laser powers or exposure times to exclude FRAP-induced structural variations, potentially affecting data in Figures 5 and 7? Moreover, FRAP ROIs were chosen to be 3–4 μm in size, occasionally resulting in ROIs larger or smaller than the exposed condensates. This size mismatch can affect the observed diffusion dynamics and the balance between internal diffusion and exchange with the surrounding dilute phase. How did the authors account for this effect in their analysis?

We apologize for omitting these parameters. We measured the laser power with a power meter on the sample plane: during FRAP experiments, the laser operated at 0.92 mW, distributed over a circular region with a fixed diameter of 5.22 μm in “tornado” mode, with an exposure time of 50 ms. The “tornado mode” is a circular scanning mode, specifically developed by Olympus, that significantly reduces laser exposure time for photobleaching applications, for which conventional raster scanning could be slow and result in inadequate bleaching. In the revised version of the manuscript, we now explicitly provide these values (Methods section, lines 873-874).

We used full laser power to induce the total and irreversible bleaching of the spot, which is essential for reliable FRAP analysis (see for example Reits et al., Nat Cell Biol 2001, doi: 10.1038/35078615; or Srikantha et al., PLoS One 2022, doi: 10.1371/journal.pone.0261925). For this reason, we did not compare fluorescence recovery using different laser powers or exposure times since such procedures are not common in the literature, where usually full laser power is used. We monitored potential laser-induced photodamage by our cells by acquiring brightfield images immediately after FRAP acquisitions: we never observed blebbing or cytoplasmic protrusions (which are signs of cellular apoptosis), thus we could exclude that the FRAP exposure was too intense for the cells. For the same reason, structural variations of the proteins of interest is unlikely to happen on such small timescales. Moreover, previous studies reporting FRAP measurements on the same proteins as in our work employed comparable laser powers, ROI sizes and exposure times (for example, G3BP1-GFP: Yang et al., Cell 2020, doi 10.1016/j.cell.2020.03.046; for FUS-GFP: Patel et al., Cell 2015, doi 10.1016/j.cell.2015.07.047; for SOD1-GFP: Brasil et al., PNAS 2019, doi 10.1073/pnas.1902483116; for TDP-43-GFP: Kitamura et al., Communications Biology 2024, doi 10.1038/s42003-024-06410-3; and many others cited in our References section) and they never report any thermal-induced damages.

Regarding ROI sizes, they were always fixed at a diameter of 5.22 micron: see methods, lines 873-874, where we already wrote: “to bleach for 50ms a round region of 30 pixels in diameter (1 px = 0.174 μm)”, and now we better specified by adding: “thus obtaining a ROI of 5.22 microns in diameter”. This parameter was never changed to allow comparison across different conditions. We verified that our conclusions were robust by comparing recovery curves from condensates of different sizes: even when their dimensions were smaller or bigger than the ROIs, the overall recovery dynamics and parameters remained consistent (see Figures R1 and R2 in the next page for two examples of SOD1 or TDP-43²²⁰: these are part of the new Supplementary Figure 7C). This suggests that size mismatches did not bias our analysis within the error of the measurement.

Figure R1: FRAP images and curves of TDP-43²²⁰ cells treated with ARS. Here we compared recovery curves from ROIs with size bigger (upper panels, orange curve in the graph) or smaller (lower panels, yellow curves in the graph) than the stress granule of interest. Resulting recovery curves looked similar. Scale bar = 10 microns.

Figure R2: FRAP images and curves of SOD1 cells (A4V mutation). Here we compared recovery curves from ROIs with size bigger (upper panels, orange curve in the graph) or smaller (lower panels, yellow curves in the graph) than the stress granule of interest. Resulting recovery curves looked similar. Scale bar = 10 microns.

We now include these important aspects in the revised version of the text (lines 474-479): “Throughout the experiments, the size of FRAP ROIs was kept constant to ensure comparability across acquisitions. As shown in Supplementary Fig. 7C, we verified the robustness of our data by comparing recovery curves from condensates of varying sizes. Even when their dimensions were smaller or larger than the FRAP ROIs, the overall recovery dynamics and parameters remained consistent, thus suggesting that variations in condensates size did not bias our analysis”.

3) As for the FRAP case, the EOM-BM characterization of biomolecular condensates presented in the manuscript is mainly structural. The study does not provide any quantified mechanical property for condensates. I believe that the terms “mechanical characterization” and “quantified mechanical properties” are overused throughout the manuscript and in the abstract. In this regard, why the authors did not extrapolate any viscosity values from the condensates dataset?

We thank the reviewer for this valuable comment, with which we partially agree. In our study we quantified differences between Brillouin shifts of the condensates: for example, TDP-43²²⁰ exhibited higher shifts than G3BP1, reflecting differences in their Longitudinal Moduli (M) and thus in their viscoelastic properties, confirmed by a strong correlation with FRAP data.

We acknowledge that our analysis does not provide absolute mechanical values of M, but rather relative differences based on Brillouin shifts. This choice reflects the fact that deriving M from Brillouin data requires precise, point-by-point knowledge of both the refractive index and the density of the condensates: these parameters are experimentally challenging to measure. Even when one of them is known, the other typically needs to be approximated using models, as done in previous Brillouin microscopy studies on biomolecular condensates (see for example Schluber et al., eLife 2022, doi: 10.7554/eLife.68490, or Beck et al., Molecular Biology of the Cell 2024, doi: 10.1091/mbc.E24-03-0128). For this reason, we considered that calculating an approximated M would not add meaningful value and that the Brillouin shift itself, widely recognized in the literature as a reliable proxy for stiffness, was sufficient to characterize our data (as deeply discussed in literature, see e.g. Bouvet et al., Nature Photonics 2025, doi: 10.1038/s41566-025-01681-6) and to describe the observed mechanical differences among condensates. For these reasons, we incorporated different internal controls such as GFP and G3BP1, and explicitly quantified their Brillouin shifts to validate our relative mechanical comparisons.

That said, we agree that the term “mechanical characterization” might not be detailed enough to reflect the scope of our results. We thank the reviewer for pointing this out. We have revised the text accordingly:

- in the abstract (line 25): “Using this system, we quantified the Brillouin shifts of several protein condensates in living cells and validated our findings with FRAP measurements”*
- in the Introduction, line 200: “To validate our Brillouin shifts measurements, we conducted parallel FRAP acquisitions:”*

- In the results, line 420: “We finally applied the EOM-BM system to investigate changes in the Brillouin shifts of biomolecular condensates implicated in amyotrophic lateral sclerosis”
- In the Discussion, line 575: “We also performed long-term measurements in live cells expressing different SGs to study their Brillouin shifts”.

For what concerns the viscosity, we attempted to extract this information by analyzing the deconvolved FWHM values of the condensates. However, these showed no significant differences between different conditions a part from G3BP1-GFP and TDP-43²²⁰-GFP (see Figure R3):

Figure R3 Brillouin FWHMs of condensates.

This is likely due to the limited precision of FWHM estimates of a Brillouin Microscope, a parameter much more difficult to obtain than the Brillouin shift: FWHM is indeed highly sensitive to the fitting procedure and to the overall quality of the data (Bouvet et al., Nature Photonics 2025). Our plan is to re-analyze such data using more advanced computational approaches (e.g., AI-assisted analysis or correlation-based algorithms that consider neighboring pixels), which we see as a future perspective of this work.

Moreover, while in the present work we could correlate Brillouin shifts with FRAP-derived immobile fractions, a similar correspondence between Brillouin FWHM (related to viscosity) and FRAP diffusion coefficient is not so straightforward. Indeed, the diffusion coefficient D is inversely dependent on local viscosity η through the Stokes-Debye-Einstein formula (i.e. $D = \frac{k_B T}{6\pi\eta R}$), which applies only to perfectly spherical particles of known radius R (Srikantha et al., *Plos One* 2022, doi: 10.1371/journal.pone.0261925): these conditions are not satisfied in our case, since proteins cannot be simply modeled as spherical particles. For these reasons, we limited our conclusions to the parameters we could robustly quantify.

MINOR:

1) Several figures have heterogeneous font size within the same panels. In some cases the text is small and it is hardly readable. Graphs within the same figures and panels can differ in size and could be better aligned/organized. FRAP ROIs highlighted by the red circles in Figure S6 are barely visible.

We apologize for this inaccuracy. In the revised version of the manuscript, we now provide figures with homogeneous fonts sizes and bigger text. We also now exported all the figures as vectorial objects with higher resolution, and show FRAP ROIs with a zoom on the desired region to better visualize the fluorescence recovery. We thank very much the reviewer for this suggestion and hope the figures readability has improved in the revised manuscript.

2) Figure 2B: why 14 points in the upper graph (calibration)? Shouldn't we expect 16 points?

The datapoints in Figure 2B are indeed 16, but the first and last pairs in the graph share the same y value because each corresponds to one Stokes and one anti-Stokes overlapping, coming from different EOM frequencies. This is also visible in Figure 2A, where the curves for $\nu_{EOM} = 7$ GHz (blue) and $\nu_{EOM} = 11.5$ GHz (violet) show overlapping peaks at the first and last positions. In Figure 2B, this resulted in a first point in $\nu_{EOM} = 7$ GHz (Stokes) and $FSR - 2\nu_{EOM} = 7$ GHz (anti-Stokes of $\nu_{EOM} = 11.5$ GHz). The last points are instead $2*\nu_{EOM} = 23$ GHz (2nd harmonic of $\nu_{EOM} = 11.5$ GHz) and $FSR - \nu_{EOM} = 23$ GHz (anti-Stokes of $\nu_{EOM} = 7$ GHz). With this consideration, the total number of points is 16 as expected. We now clarified this aspect in the revised caption of Figure 2 (lines 1226-1227): “The first and last pairs of datapoints overlapped, as they corresponded to Stokes and Anti-Stokes signals arising from different EOM fundamental frequencies”.*

3) Literature 1 (intro): I share with the authors the emergency of developing instruments capable of investigating "in vivo 3D biological environments" with high spatial and temporal resolution. Their development fits well within this context, but the authors did not discuss efficient and faster techniques: in vivo imaging with fluorescence lifetime probes can quantify membrane tension (e.g. doi: 10.1021/jacs.8b13189), or local viscosity (e.g. Bodipy probes).

We thank the reviewer for highlighting these important techniques, which allowed us to expand the introduction of our paper with alternative methods for biomechanical characterizations of cells.

Mechanosensitive fluorescent probes, such as Flipper-TR, are designed to emit fluorescence in response to changes in plasma membrane tension caused by external forces, rather than directly reporting the intrinsic stiffness or mechanical properties of cellular structures. These probes are not label-free, requiring chemical incorporation into cells, and their staining is limited to specific organelles or membrane regions, which constrains their applicability for global or long-term measurements.

*BODIPY-based molecular rotors are bright, stable, and tunable fluorophores whose emission can reflect local viscosity. However, their fluorescence is not exclusively viscosity-dependent as it can be influenced by solvent polarity, molecular aggregation, or interactions with proteins and lipids (Paez-Perez et al., *Angewandte Chemie* 2023, doi: 10.1002/anie.202311233; Kowada et al., *Chem. Soc. Rev.* 2015, doi: 10.1039/c5cs00030k). Additional phenomena such as intersystem crossing or aggregation-induced quenching can complicate interpretation. Both approaches require incubation of the dye in cells or conjugation to targeting molecules, unlike genetically encoded markers such as GFP.*

In contrast, Brillouin microscopy is fully label-free and can generate mechanical maps over genetically encoded proteins of interest (as we did in our manuscript), enabling direct, non-invasive measurements of intracellular condensates.

We have now included a discussion of these complementary approaches in the revised Introduction (lines 78-87): "Other techniques capable of in vivo capacities, such as Mechanosensitive fluorescent probes²⁴, report fluorescence changes in response to alterations in plasma membrane tension caused by external forces: consequently, they do not directly reflect the intrinsic mechanical properties of cellular structures. Similarly, tunable fluorophores like BODIPY-based molecular rotors are probes whose emission correlates with local viscosity, though it can also be affected by solvent polarity, molecular aggregation or interactions with proteins and lipids^{25,26}. Importantly, these probes are not label-free and require chemical incorporation into cells; moreover, their staining is confined

to specific organelles or membrane regions, limiting their use in global or long-term mechanical measurements”.

4) Literature 2 (intro): the authors did not discuss AFM, which remains the principal technique for characterizing the mechanics of liquid–solid phase transitions in biomolecular condensates, yet in vitro. AFM, like Brillouin microscopy, is label-free and complementary, as it characterizes mechanical properties at high temporal resolution, as in microrheology, quantifying Young’s moduli, surface tensions, and viscosities (doi: 10.1016/j.xcrp.2025.102430, and 10.1073/pnas.2304714120).

As correctly noted, AFM is a label-free technique for directly measuring mechanical responses, as it is sensitive to the Young’s modulus and can quantify properties such as surface tension and viscosity. However, AFM is generally limited to in vitro samples, as it cannot probe living cells or within intact tissues, and it lacks 3D capacity. In contrast, Brillouin microscopy is also label-free but offers the unique advantage of being compatible with living cells and tissues, enabling non-invasive, 3D mechanical mapping in both in vitro (Beck et al., Molecular Biology of the Cell 2024, doi: 10.1091/mbc.E24-03-0128) and in vivo settings, as demonstrated in our study.

We thank the reviewer for the opportunity to expand our Introduction with experimental techniques complementary to Brillouin Microscopy. We have now included this discussion in the revised Introduction to clarify the complementary roles of AFM and Brillouin microscopy, highlighting how our approach extends mechanical characterization into living cellular environments that are inaccessible to AFM (lines 71-77): “AFM is considered the main technique for measuring the Young’s Modulus E , whereby the material experiences a change in volume, through nano-indentation of the sample’s surface. While it has been applied to biomolecular condensates in vitro^{21,22}, AFM lacks 3D capacities as its measurements are averaged along the axial direction and strongly depend on mathematical models to extract E . AFM spatial resolution is also limited: despite the use of a very small tip, its resolution on biological materials generally exceeds several fold the tip size²³”.

Reviewer #4:

This manuscript presents the results of a study conducted on stress granules (SGs) in living cells using a Brillouin microscope stabilized using an electro-optic modulator (EOM). The use of EOMs in Brillouin spectrometers has previously been demonstrated (ref. 36), however not in VIPA-Brillouin systems to the best of my knowledge. So there is some novelty in this application. The results are notable, especially in the presence of large temperature fluctuations (~2.5°C; Supplementary Information).

We thank the reviewer for appreciating the impact and the novelty of our manuscript.

Comments/questions:

- it is not clear how the deconvolution of the Brillouin spectra is performed (Brillouin Spectral Fitting) between having a Lorentzian-shape Brillouin signal and a Gaussian EOM signal.

We apologize if this point was not clearly written. We performed a fit of the Brillouin spectrum using the convolution between two functions: (i) the sum of two Lorentzian functions (for the Brillouin peaks) and two Gaussian functions (for the EOM peaks), and (ii) the experimental VIPA lineshape, extracted during the EOM calibration step. In this way, we obtained directly the Brillouin shift and width from the output parameters of the non-linear fitting procedure.

We included this part in the revised text (lines 766-770): “We fitted Brillouin signals by using the convolution between two functions: (i) the sum of two Lorentzian functions (for the Brillouin peaks) and two Gaussian functions (for the EOM peaks), and (ii) the experimental spectrometer line shape, extracted during the EOM calibration step. In this way, retrieved Brillouin FWHMs were not affected by VIPAs broadening.”

- it is not clear what is the composition of the SGs - what if those are lipid droplets or else? How specific is the staining for validation?

SGs are membrane-less cytoplasmic compartments that are physiologically formed by cells in response to stressors such as oxygen reactive species, heat shock or toxin exposure. Their composition is made by a network of protein–protein, protein–RNA, and RNA–RNA interactions (Alberti et al., Nature Reviews Molecular Cell Biology 2021), so they are not made by lipid droplets. For what concerns the staining, we confirm that the GFP signal is

highly specific for the protein of interest: indeed, we permanently inserted into the cell's genome a single vector encoding for a chimeric construct, made by the protein of interest fused to a GFP tag.

We thank the reviewer for pointing out this aspect, that is important for our study and helps improve the comprehension of our results for an interdisciplinary audience. In the Introduction of the revised text, we now provide a comprehensive explanation of SGs composition (lines 47-48: "They consist of a dynamic network of protein–protein, protein–RNA, and RNA–RNA interactions⁴") and highlight the specificity of the staining for the different proteins (line 444-445: "thus, the GFP signal was highly specific for the protein of interest").

- how generalisable are these results to other proteinopathies?

These results prove that our improved Brillouin Microscope reliably quantify the mechanical properties of protein condensates in living and fixed cells and that it even outperforms already established techniques such as FRAP. Because Brillouin microscopy is label-free and noninvasive, it could be applied to different membrane-less organelles and protein condensates implicated in diverse proteinopathies (e.g., Huntington's, Parkinson's, Alzheimer's Diseases), irrespective to their molecular composition. The generalizability to other disease-relevant condensates will depend on their size, localization, and optical accessibility within cells. Thus, while our present data provide proof of concept, the method offers a broadly applicable framework for future studies across multiple proteinopathies.

We thank the reviewer for asking this question, giving us the opportunity to further discuss the generalizability of our results. In the final Discussion section of the revised version of the manuscript, we now provide a comment on this (lines 606-610): "Beyond the ALS and FTD cases examined here, the label-free and noninvasive nature of Brillouin microscopy could be applied to other membrane-less organelles and protein condensates implicated in diverse proteinopathies (e.g., Huntington's, Parkinson's, Alzheimer's Diseases), regardless of their molecular composition".

- how translatable are these findings to medical applications of Brillouin microscopy?

Our work establishes a technological advance that improves the robustness and precision of Brillouin microscopy in live-cell measurements, opening the way to long time-lapse biomechanical measurements that were so far hard to obtain.

While the present study is at a fundamental level, focusing on the biophysics of protein condensates, the improvements we introduce address key limitations that have so far hindered a broader use of Brillouin microscopy in biomedical contexts. These features allow the technique to be more suitable for large-scale studies, including applications in disease modeling and potentially in clinical diagnostics. However, translation to medical applications will still require validation in more complex systems (e.g., tissue models or patient-derived samples), as well as integration with complementary clinical imaging modalities. In this context, our data demonstrate that Brillouin Microscopy can be applied to the characterization of biomolecular condensates in fixed samples and that cell fixation does not affect the mechanical properties of condensates. This is a potential powerful medical application as it paves the way to mechanical characterizations of patient-derived post-mortem tissues or for screening selective chemical compounds for their ability to disrupt pathogenic condensates. In this sense, our findings provide an enabling step toward medical applications, by solving a critical limitation in instrument stability and reproducibility.

We thank the reviewer for the opportunity to discuss about this important point: in the final part of our revised manuscript (lines 599-602), we now briefly discuss about medical applications of our work: “First, it can be applied to the characterization of biomolecular condensates in fixed samples: this capability would be of particular relevance for medical applications regarding the mechanical characterizations of patient-derived post-mortem tissues.”.

Response to Reviewers

We thank the reviewers for the time devoted to the review and for their positive evaluation of our manuscript. We are pleased that the revisions have successfully addressed all comments.

Please find attached a detailed, point-by-point response to all reviewer comments. Reviewer remarks are presented in bold, with our responses in italics.

Reviewer #1:

All my comments have been properly addressed. I am satisfied with the authors's replies. I recommend publication of the manuscript in its present form.

We appreciate this comment and are grateful to the reviewer for recommending the publication of our manuscript, surely improved after his/her suggestions.

Reviewer #2:

We appreciate the Reviewer's positive assessment and trust that the manuscript is now suitable for acceptance.

Reviewer #3:

The revised manuscript is now improved and is essentially ready for publication.

I am still convinced that in some parts of this manuscript the mechanical properties are described in terms that lies between qualitative and quantitative description. However, after the authors' corrections, this is now only a semantic issue, beyond the scope of the manuscript.

Last: the statement “AFM spatial resolution is also limited: despite the use of a very small tip, its resolution on biological materials generally exceeds several times the tip size” is fundamentally incorrect. Hundreds of publications demonstrate that AFM resolution on biological materials can be comparable to or even better than the probe size, ranging from biological membranes (doi: 10.1073/pnas.0437992100) to biomolecular condensates in vitro (doi: 10.1038/s41586-020-1977-6). I am providing these dois as examples, not as suggestions for inclusion, obviously. The authors' claim applies only to images acquired at the apical membrane of some eukaryotic cells, where, in any case, AFM resolution does not exceed several times the tip size.

That said, I confirm that the manuscript is very interesting, clearly written and organized.

We are grateful to the reviewer for acknowledging the relevance and the qualities of our manuscript.

We also thank him/her for the careful reading. We understand that the statement regarding AFM spatial resolution could be misleading. In line with the journal's request to shorten the Introduction, we relocated the section discussing complementary biomechanical techniques to the beginning of the Discussion (new lines 515–544). Here, we have removed the said AFM sentence from the text to avoid any incorrect or generalized claims. We believe that our response to this minor point will help improve the clarity and quality of the manuscript, and make it suitable for acceptance.